# KRAS G12V neoantigen specific T cell receptor for adoptive T cell therapy against tumors

Dan Lu[1,2,3,8], Yuan Chen[1,2,4,8], Min Jiang[1,2,8], Jie Wang[1,2], Yiting Li[1,2], Keke Ma[1,2,5], Wenqiao Sun[1], Xing Zheng[1], Jianxun Qi ⓘ[1], Wenjing Jin[6], Yu Chen[6], Yan Chai[1], Catherine W. H. Zhang[6], Hao Liang[4], Shuguang Tan ⓘ[1,2,5,7] ✉ & George F. Gao ⓘ[1,2,7] ✉

KRAS mutations are broadly recognized as promising targets for tumor therapy. T cell receptors (TCRs) can specifically recognize KRAS mutant neoantigens presented by human lymphocyte antigen (HLA) and mediate T cell responses to eliminate tumor cells. In the present study, we identify two TCRs specific for the 9-mer KRAS-G12V mutant neoantigen in the context of HLA-A*11:01. The TCR-T cells are constructed and display cytokine secretion and cytotoxicity upon co-culturing with varied tumor cells expressing the KRAS-G12V mutation. Moreover, 1-2C TCR-T cells show anti-tumor activity in preclinical models in female mice. The 9-mer KRAS-G12V mutant peptide exhibits a distinct conformation from the 9-mer wildtype peptide and its 10-mer counterparts. Specific recognition of the G12V mutant by TCR depends both on distinct conformation from wildtype peptide and on direct interaction with residues from TCRs. Our study reveals the mechanisms of presentation and TCR recognition of KRAS-G12V mutant peptide and describes TCRs with therapeutic potency for tumor immunotherapy.

Kirsten Rat Sarcoma Viral Oncogene Homolog (*KRAS*) is the most frequently mutated gene in multiple tumors, including non-small-cell lung cancer (NSCLC) (China, ~12%), colorectal cancer (China, ~46%), and pancreatic cancer (China, ~90%)[1,2]. KRAS mutations, especially the most common mutations at codons 12 and 13, promote tumor cell metabolic reprogramming, cellular proliferation and survival[3,4]. The top three most frequent KRAS codon 12 mutations are G12D, G12V, and G12C among all tumor types[2]. Clinical investigations have revealed that KRAS mutations are associated with poor outcomes[5].

The KRAS protein functions as a GTPase, cycling between a GTP loaded "ON" state and a GDP-loaded "OFF" state[6,7]. KRAS mutations can impair the GTP hydrolysis activity and lock the protein in the GTP-loaded "ON" state, resulting in enhanced KRAS mediated downstream effector pathways, including the mitogen-activated protein kinase (MAPK) and phosphatidylinositol 3-kinase (PI3K) pathways[7,8]. Tumor cells carrying KRAS mutations are endowed with cellular proliferation and survival, and they accumulate in tumor patients. KRAS mutations are usually mutually exclusive, and a previous study revealed that only 304 specimens (from 263 patients) out of a cohort of 8750 KRAS-mutant tumors (from 7790 patients) had multiple RAS mutations[9]. Therefore, KRAS mutations are broadly recognized as promising targets for tumor therapy[2,7].

[1]CAS Key Laboratory of Pathogen Microbiology and Immunology, Institute of Microbiology, Chinese Academy of Sciences, Beijing, China. [2]Savaid Medical School, University of Chinese Academy of Sciences, Beijing, China. [3]Department of Immunology, Beijing Children's Hospital, Capital Medical University, National Centre for Children's Health, Beijing, China. [4]Collaborative Innovation Centre of Regenerative Medicine and Medical BioResource Development and Application Co-constructed by the Province and Ministry, Guangxi Medical University, Nanning, Guangxi, China. [5]Shenzhen Children's Hospital, Shenzhen, Guangdong, China. [6]YKimmu (Beijing) Biotechnology Co., Ltd, Beijing, China. [7]Beijing Life Science Academy, Beijing, China. [8]These authors contributed equally: Dan Lu, Yuan Chen, Min Jiang. ✉e-mail: tansg@im.ac.cn; gaof@im.ac.cn

However, due to the featureless structure of the KRAS protein and the lack of an adequate binding pocket for small molecules, KRAS has been regarded as an "undruggable" target since its identification 40 years ago[10]. In 2021, the United States Food and Drug Administration (US FDA) accelerated the approval of sotorasib, a small molecule drug that covalently binds to cysteine 12 in the GDP-bound state of the KRAS-G12C mutant, to treat advanced-stage KRAS-G12C mutant NSCLC and marked a breakthrough in tumor therapy[11]. However, developing small molecules to target other KRAS mutations remains challenging[2].

T cell receptors (TCRs) are highly sensitive in responding to intracellular antigens presented by human leukocyte antigen (HLA) molecules on the cell surface and are broadly recognized as prospective strategies in the development of therapeutics for solid tumors[12–14]. The approval of the TCR drug KIMMTRAK (tebentafusp) for unresectable or metastatic uveal melanoma by the US FDA in 2022 has initiated an era for the development of TCR based drugs for tumor immunotherapy[15,16]. Intracellular mutated KRAS proteins can be processed and presented by the major histocompatibility complex (MHC, or the human leukocyte antigen [HLA]) as a "non-self" neoantigen, and subsequently be recognized by specific TCRs to elicit T cell immune responses[17]. T cells specific for KRAS mutants would eliminate tumor cells with KRAS mutations through secreted cytokines or via direct killing with granzyme B and perforin.

In 2016, Tran et al. reported for the first time the adoptive transfer of in vitro expanded HLA-C*08:02 restricted tumor-infiltrating lymphocytes, which contain large amounts of KRAS-G12D mutant specific T cells, in a patient with metastatic colorectal cancer, demonstrating effective anti-tumor efficacy[18]. They subsequently proved that infusion of TCR-engineered T (TCR-T) cells targeting KRAS-G12D in the context of HLA-C*08:02 resulted in regression of metastases in a patient with pancreatic cancer[12]. These clinical investigations provide solid evidence that TCR-T cells targeting tumor-intrinsic KRAS mutants could be a promising strategy for the treatment of solid tumors.

Multiple studies demonstrate that the KRAS G12 mutation is a "hotspot" region for HLA class I (HLA-I) restricted T cell epitopes[17,19]. Multiple HLA-I restricted T cell epitopes have been identified, including HLA-A*11:01 (G12D and G12V), -A*03:01 (G12V), -B*07:02 (G12R), -C*01:02 (G12V), -C*08:01 (G12D), and -C*05:01 (G12D), and cognate TCRs are also reported to be specific for these epitopes[18,20–24]. Targeted mass spectrometry analysis identified that the mutated peptides can also be processed and presented by HLA-A*30:01, -A*68:01, -C*03:03 and -C*03:04[19]. HLA-A*11:01 is highly prevalent in East Asia with a gene frequency of 21%[25]. Two overlapped HLA-A*11:01 restricted epitopes have been identified for KRAS-G12 mutants, i.e., the 9-mer peptide and the N-terminal extended 10-mer peptide for G12V or G12D mutants[20–22]. Though overlapping with each other, T cells responsive to 9-mer or 10-mer epitopes do not display cross responses to one another[17].

The structural bases for peptide presentation and TCR recognition are important both for our understanding of immunogenicity of the epitopes and beneficial for directed evolution of high affinity TCRs or de novo design of molecules targeting specific epitopes[16,26–28]. The structures of 9-mer (GADGVGKSA) or 10-mer (GADGVGKSAL) KRAS-G12D mutant peptides restricted by HLA-C*08:02 and the cognate TCR complex have been reported, revealing distinct presentation and recognition mechanisms of these overlapping peptides[29]. The structure of the 10-mer KRAS-G12D epitope restricted by HLA-A*11:01 was determined and the recognition mechanisms by specific TCRs were also elucidated[22]. However, the mechanisms of presentation and TCR recognition of the 9-mer KRAS-G12 mutant peptide in the context of HLA-A*11:01 remains unknown.

In the present study, we identify two TCRs specific for the 9-mer KRAS-G12V mutant peptide (KRAS-G12V-9) by immunization of HLA-A*11:01 transgenic mice. Of note, one of the TCRs, 1-2C, is a public TCR identified in multiple mice. Chimeric TCR-T cells are constructed and show specific responses against multiple tumor cells with the KRAS-G12V mutation. 1−2C TCR-T cells show tumor suppression efficacy in a tumor-bearing mouse model and improved inhibition of tumor growth in combination with anti-PD-1 antibodies. The structures of the TCR complex with the 9-mer KRAS-G12V mutant peptide presented by HLA-A*11:01 are determined. Structural analyses reveal distinct presentation mechanisms of the 9-mer peptide from the 10-mer peptide, providing the structural basis for specific recognition of the G12V mutation by TCRs. These findings provide fundamental information for our understanding of the two distinct epitopes of the KRAS-G12V mutant and will be of benefit for the future design of biologics targeting KRAS mutant tumors.

## Results
### Identification of KRAS-G12V specific TCRs from HLA-A*11:01 transgenic mice
This study aimed to identify KRAS mutant specific TCRs from HLA-A*11:01 transgenic mice and investigate mechanisms of peptide presentation and TCR recognition of KRAS mutant epitopes in the context of HLA-A*11:01 (Supplementary Fig. S1). HLA-A*11:01 transgenic mice were subcutaneously immunized with the KRAS-G12V-9 peptide (VVGAVGVGK) and the spleen cells from nine out of 12 mice displayed specific responses against the KRAS-G12V-9 peptide (Supplementary Fig. S2a). Spleen cells from four mice (Mus-T5, Mus-TF1, Mus-TF2, and Mus-TF6) that showed substantial KRAS-G12V-9 specific responses were stained with KRAS-G12V-9/HLA-A*11:01 tetramer, and the tetramer positive CD8⁺ T cells were sorted by single-cell sorting with flow cytometry (Supplementary Fig. S2b, c). Variable (V) genes of the paired TCR α and β chains in each of the specific T cells were amplified by 5' RACE and subsequently sequenced with a previously established procedure[30,31].

The analysis of the KRAS-G12V-9 specific TCR repertoires from the four mice revealed that the most frequent TCRs identified in Mus-T5, Mus-TF1, Mus-TF2 and Mus-TF6 were 1-2C [14/15 (93%)], 1-2C [6/14 (43%)], A13B13 (TRAV13-5*01, TRBV13-2*01) [5/19 (19%)] and 3-2E (TRAV8-1*03, TRBV16*01) [8/25 (32%)] (Fig. 1a; Supplementary Table S1). Of note, 1-2C TCR (TRAV7-2*02, TRBV13-2*01) is a public TCR that presents in each of the four mice and dominated KRAS-G12V-9 specific TCRs in Mus-T5 and Mus-TF1 (Fig. 1a).

To validate the binding capability and functional potency of the identified TCRs, chimeric TCRs were constructed with the constant domains of both the TCR α and β chains replaced with human counterparts (TRAC and TRBC1), whereas murine variable domains were preserved. HEK-293T cells were co-transfected with plasmids expressing chimeric TCR pairs from seven clones and human CD3-CD8. The expression of TCRs and the binding capacities to KRAS-G12V-9/HLA-A*11:01 tetramer were analyzed by flow cytometry. The expression of TCRs was detected by staining with antibodies specific to the constant domain of the human αβ TCR and the expression of seven TCRs could be observed with varied frequencies from 29.0 to 50.30 % (Fig. 1b; Supplementary Fig. S3). Four TCRs demonstrated specific binding to KRAS-G12V-9 mutant peptide/HLA (pHLA) tetramers, but no binding to 9-mer KRAS-G12 wildtype (KRAS-G12wt-9) pHLA tetramers was observed (Fig. 1c, d). 1-2C (21.30%) and 3-2E (17.50%) TCRs demonstrated a substantially higher frequency of KRAS-G12V-9/HLA-A*11:01 tetramer staining positive cells than the other TCRs and were therefore selected for further binding and functional investigations (Fig. 1d).

### Profiles of the 1−2C and 3-2E TCRs binding to KRAS mutants
The binding specificity to KRAS mutants is a critical issue for the safety of TCR-T cell therapy. To determine the binding specificity of the 1-2C and 3-2E TCRs, HEK-293T cells transiently expressing the 1-2C or 3-2E TCR were stained with pHLA tetramers loaded with varied KRAS mutant peptides and analyzed by flow cytometry (Fig. 2a). The peptides loaded in pHLA tetramers include KRAS-G12wt-9, KRAS-G12V-9

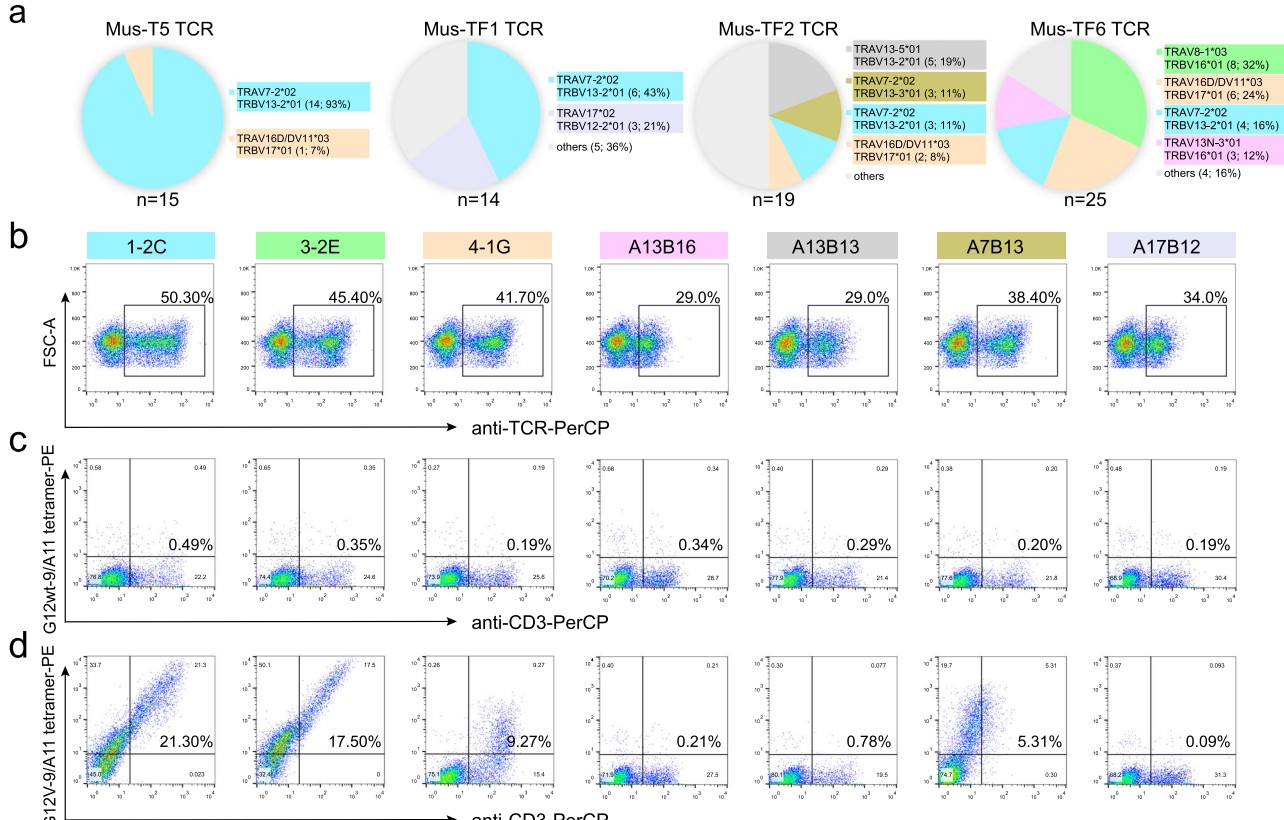

**Fig. 1 | KRAS-G12V-9 specific TCR screening and binding validation in HEK-293T cells. a** TCR repertoire analysis of cloned KRAS-G12V-9/HLA-A*11:01 tetramer⁺ CD8⁺ T cells from Mus-T5, Mus-TF1, Mus-TF2, and Mus-TF6. Numbers below the pie charts represent the number of TCRs identified in the mouse. **b** The expression of TCRs in HEK-293T cells after co-transfection of chimeric TCR and human CD3-CD8 constructs was detected by staining with antibodies specific to human αβ TCR. Binding of wildtype KRAS-G12wt-9/HLA-A*11:01 tetramer (**c**) or mutated KRAS-G12V-9/HLA-A*11:01 tetramer (**d**) to HEK-293T cells transiently expressing the indicated TCRs as in (**b**) for each panel. The Y axis represents the staining events by the indicated tetramer, while the X axis represents the staining events of CD3-positive cells by the anti-CD3 antibody. Data in (**b**), (**c**), and (**d**) are representative of three independent experiments.

and a subset of KRAS codon 12 and 13 mutant peptides (KRAS-G12C-9, KRAS-G12D-9, KRAS-G12A-9, KRAS-G12S-9, KRAS-G12R-9, or KRAS-G13D-9) frequently observed in tumor specimens (Supplementary Table S2). All the mutant peptides could properly bind the HLA-A*11:01 molecule and form stable pHLA complex proteins (Supplementary Fig. S4). The results demonstrated that the 1-2C and 3-2E TCRs exhibited specific binding to the G12V pHLA tetramer and showed weak-binding to the KRAS-G12C-9 mutant pHLA tetramer, whereas no binding was observed to wildtype or other KRAS mutant pHLA tetramers (Fig. 2b).

To further characterize the binding profiles of the 1-2C and 3-2E TCRs to varied KRAS mutants in the context of HLA-A*11:01, protein-based surface plasmon resonance (SPR) binding assays were conducted with soluble TCR and HLA proteins. Recombinant HLA proteins of HLA-A*11:01 with wildtype or mutant peptides were captured on streptavidin sensor chips, and serial dilutions of recombinant 1-2C and 3-2E TCR proteins were flowed through as analytes. The results showed that 1-2C and 3-2E bind to KRAS-G12V-9/HLA-A*11:01 with KD values of $14.0 \pm 0.8\,\mu M$ and $28.0 \pm 1.9\,\mu M$, respectively, falling within typical TCR-pHLA affinity range (Fig. 2c, d)[32,33]. Consistent with the weak binding to the G12C mutant observed in cell-based binding assay, 1-2C and 3-2E displayed substantially lower, but still detectable, binding affinities for the KRAS-G12C-9 mutant pHLA, with KD values of $131 \pm 13.2\,\mu M$ and $42.5 \pm 2.5\,\mu M$, respectively. Apart from this, neither 1-2C nor 3-2E could bind to wildtype or other KRAS mutants. These results indicate that the 1-2C and 3-2E TCRs can specifically bind to the KRAS-G12V-9 mutant and may cross-recognize KRAS-G12C-9 with weak binding capacity in the context of HLA-A*11:01[17].

## Immune responses of 1-2C and 3-2E TCR-T cells

To investigate the immune responsive potency and specificity of the 1-2C and 3-2E TCRs, Jurkat cells and primary T cells were genetically engineered with lentiviruses to express chimeric 1-2C or 3-2E TCR. The immune responses of TCR-engineered Jurkat cells or primary T cells were detected using IL-2 or IFN-γ ELISPOT or ELISA assays after co-culturing with target cells loaded with peptides or tumor cells intrinsically expressing KRAS mutants.

Jurkat cells expressing the 1-2C or 3-2E TCRs were tested as effector cells, whereas K562-HLA-A11 cells, which were transduced to express HLA-A*11:01, were loaded with varied peptides and co-cultured as target cells (Fig. 3a). Dose-dependent responses, as detected by IL-2 secretion using ELISA, was observed for Jurkat cells expressing 1-2C or 3-2E TCRs upon stimulation with serial dilutions of KRAS-G12V-9 peptides co-cultured with K562-HLA-A11 cells, with a half maximal effective concentration ($EC_{50}$) of 760.4 nM for 1-2C and 498.0 nM for 3-2E (Fig. 3b, c). In contrast, no responses were observed for wildtype, G12D, or G12C peptides.

To evaluate the clinical application potency of the 1-2C and 3-2E TCRs, primary T cells were isolated from healthy donors and transduced with lentiviruses carrying the 1-2C or 3-2E TCR gene to generate 1-2C or 3-2E TCR-T cells. TCR-T cells were co-cultured with PANC-1 cells, which naturally express HLA-A*11:01, and loaded with varied peptides (Fig. 3d). The levels of IFN-γ released in supernatants after co-culturing were subsequently detected with ELISA. The results demonstrated that substantial IFN-γ secretion was induced upon specific stimulation with the KRAS-G12V-9 peptide for 1-2C and 3-2E TCR-T cells (Fig. 3e, f). Indeed, 1-2C and 3-2E TCR-

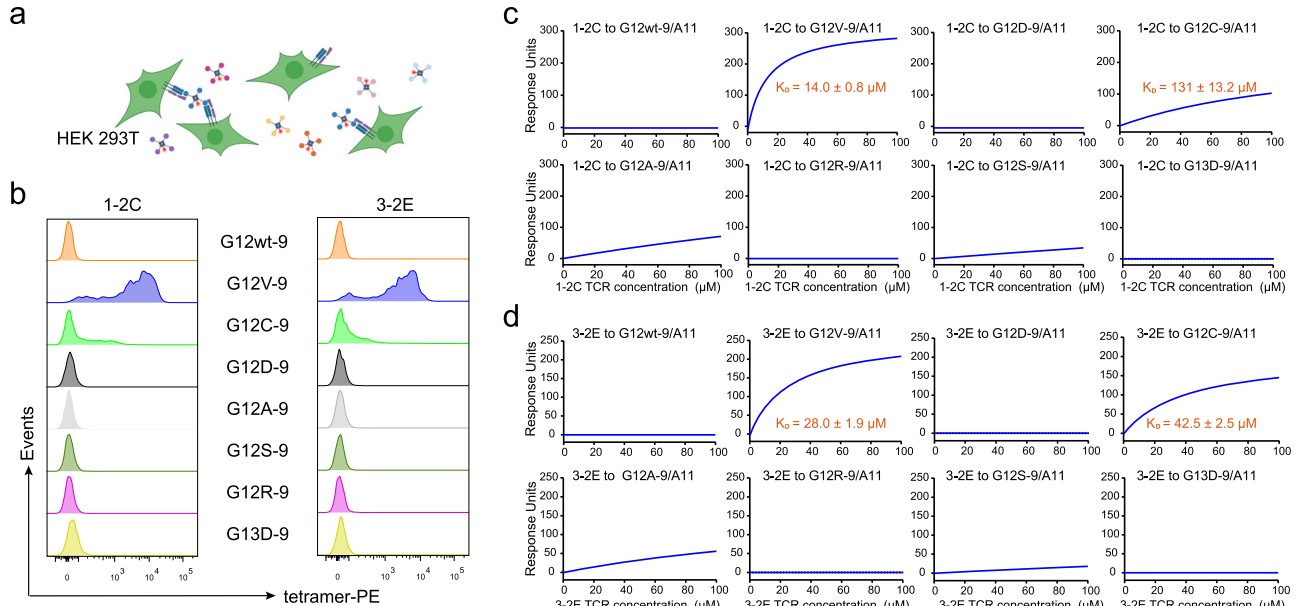

**Fig. 2 | Specific binding of the 1-2C and 3-2E TCRs to the KRAS-G12V mutant.**
**a** Schematic of the binding assay in HEK-293T cells was created with BioRender.com. **b** Binding of pHLA tetramers loaded with wildtype or varied KRAS-G12 mutant peptides to the 1-2C or 3-2E TCR expressed on HEK-293T cells was analyzed by flow cytometry. The data shown are from one of three independent experiments. SPR assay characterization of the binding profiles of 1-2C (**c**) or 3-2E (**d**) with different KRAS-G12 mutant peptide pHLA proteins. The pHLA proteins were immobilized on the chip and serial dilutions of 1-2C or 3-2E TCR proteins were then flowed through. The figures represent measurements at equilibrium with serial 2-fold dilutions of 1-2C or 3-2E proteins with concentrations ranging from 100 to 6.25 μM. The mean value of the KD was recorded after repeating each experiment three times. Source data are provided as a Source Data file.

T cells showed comparable responsive potency against KRAS-G12V-9, with $EC_{50}$ of 599.3 nM and 497.4 nM, respectively. Flow cytometry based on intracellular staining of IFN-γ revealed that the responsive TCR-T cells that secrete IFN-γ were primarily $CD8^+$ T cells (Supplementary Fig. S5). Further, we also performed CD4/8 depletion assays with IFN-γ ELISPOT analysis and found that no substantial responses could be observed for either 1-2C or 3-2E TCR-T cells when $CD8^+$ TCR-T cells were depleted (Supplementary Fig. S5c and d). Of note, no responses to G12C mutant peptide were observed for 1-2C or 3-2E TCR-T cells, indicating that the decreased binding affinity to KRAS-G12C-9 pHLA impaired the T cell responsiveness of the TCRs.

The responses of 1-2C and 3-2E TCR-T cells were further tested against varied tumor cells carrying mutant KRAS genes, instead of target cells exogenously loaded with peptides. PANC-1 cells, which intrinsically express HLA-A*11:01, were genetically engineered by lentiviral transduction to stably express wildtype or varied KRAS-G12 mutants, including G12V, G12D, and G12C. SW-620 and CFPAC-1 cells, which intrinsically express the KRAS-G12V mutant, were engineered to express HLA-A*11:01 by lentiviral transduction. The forced expression of HLA-A*11:01 and KRAS-G12V in target cells were confirmed by flow cytometry (Supplementary Fig. S6). The responses were investigated with 1-2C and 3-2E TCR-T cells prepared with primary T cells from three healthy donors (Supplementary Fig. S7). We found that 1-2C and 3-2E TCR-T cells could specifically respond to PANC-1 cells stably expressing the KRAS-G12V mutant, whereas no responses were observed against PANC-1 cells carrying wildtype or other KRAS mutants (Fig. 3g, j). Specific responses of 1-2C and 3-2E TCR-T cells against SW-620 or CFPAC-1 cells could only be detected for those stably expressing HLA-A*11:01 (Fig. 3h, i, k, l). Bioluminescence-based cytotoxicity assay revealed that substantial killing of PANC-1-G12V, CFPAC-1-A11 and SW-620-A11 cells could be observed for both 1-2C and 3-2E TCR-T cells (Fig. 3m–r; Supplementary Fig. S8). Taken together, these data indicate that 1-2C and 3-2E TCR-T cells specifically recognized and responded to eliminate tumor cells with KRAS-G12V in the context of HLA-A*11:01.

**Responsive specificity of the 1-2C and 3-2E TCRs**
Previous studies report that off-target binding to homologous self-peptides in the human genome may induce lethal cytotoxicity. Adoptive transfer of TCR-T cells targeting MAGE-A3/HLA-A*01:01 is reported to exhibit substantial off-target toxicity through binding to a Titin-derived peptide presented by HLA-A*01:01[34]. The TCRs identified in the present study were from HLA-A*11:01 transgenic mice, and potential concerns for off-target toxicity to self-antigens in human genome cannot be excluded. Cross reactivities to a combinatorial peptide library and homologous peptides in the human genome were therefore investigated.

A combinatorial peptide library was synthesized with each of the residues substituted with 20 amino acids (Fig. 4a). The responses of 1-2C and 3-2E TCR-T cells constructed from two donors were tested with IFN-γ-ELISA (Fig. 4b–s; Supplementary Fig. S9). We observed that the responses against G3 and G6 were highly specific that no substantial responses occurred when substituted with other amino acids. Many of the substitutions at V1 and V2 were tolerated for 1-2C TCR-T cells, as was also observed for the substitutions at V2 and G8 for 3-2E TCR-T cells. Cross reactivity at the C-terminal anchoring K9 could only be observed against a substitution with Arg for both of the TCRs, which is in line with the binding motif of HLA-A*11:01 restricted peptides. For G12V mutant position substitutions, substantially decreased responses for 1-2C TCR-T cells were noted against substitutions with Ile, Met and Pro, whereas no responses could be observed with other substitutions. Meanwhile, responses of 3-2E TCR-T cells were observed for Met and His at G12V mutant position. Overall, the responsive profiles at V2, G3, A4, V5, G6, V7, and K9 were similar for both 1-2C and 3-2E, whereas responses at V1 and G8 substantially varied for these two TCRs. We further asked whether homologous peptides exist in the human genome for the site-substituted peptides that showed cross-reactivity. However, through BLAST analyses (https://blast.ncbi.nlm.nih.gov/Blast.cgi), none of these peptides were found in the human genome. We further tried to use ScanProsite (https://prosite.expasy.org/scanprosite/) to analyze homologous peptides that meet the responsive motifs of 1-2C or 3-2E TCRs observed from peptide combinatorial

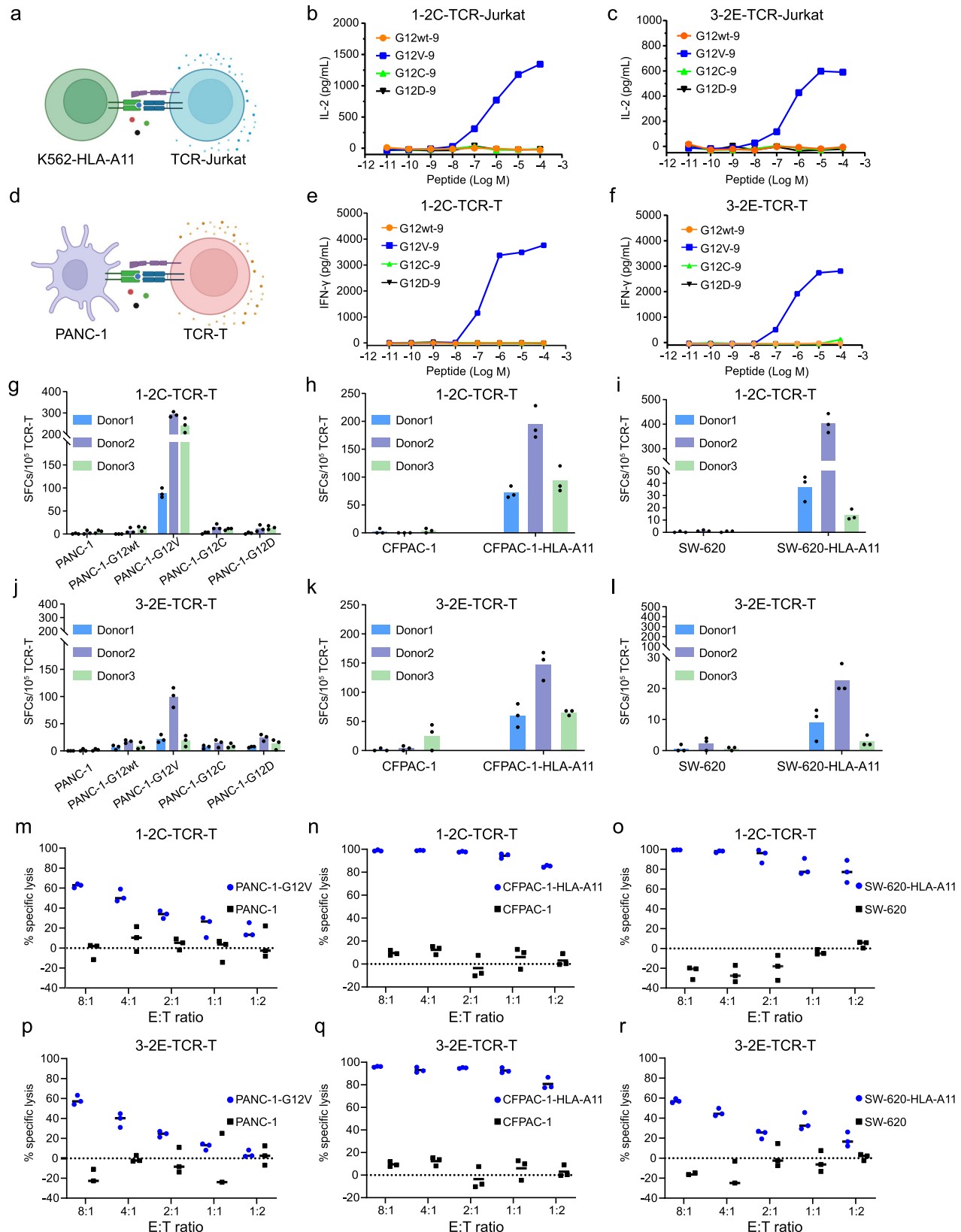

library analysis. Three homologous peptides (UniProt: Q6H795, RWGAVGVGR; UniProt: 72WD6, IFGSVGVGK; Uniprot: P9WFD4, CVGSVGIGR) were identified for 1-2C TCR recognition motif, whereas two homologous peptides (UniProt: Q2GC58, VSGCVGVFR; UniProt: A0PXA8, ITGAVGIAK) were identified for 3-2E TCR. However, none of these homologous peptides were from human genome.

We further examined the cross-reactivity of 1-2C and 3-2E TCR-T cells to homologous peptides in the human genome that share more than six out of the nine residues with the KRAS-G12V-9 peptide. Through BLAST analyses (https://blast.ncbi.nlm.nih.gov/Blast.cgi), 16 peptides were selected from 14 proteins for cross-reactivity analyses (Supplementary Table S3). The 1-2C and 3-2E TCR-T cells from three

**Fig. 3 | Antigen sensitivity and KRAS-G12V specific tumor responsiveness of 1-2C and 3-2E TCR-T cells. a** Schematic of the functional evaluation assay of the TCRs in Jurkat T cells, created with BioRender.com. Jurkat T cells were transduced to express 1-2C (**b**) or 3-2E (**c**) engineered Jurkat T cells were subsequently co-cultured with K562-HLA-A11 target cells and serially diluted peptides for 24 h. Co-cultured supernatants were analyzed by ELISA for secreted IL-2. **d** Schematic of the functional evaluation assay of the TCR-T cells engineered with primary T cells, created with BioRender.com. **e, f** 1-2C or 3-2E TCR-T cells were co-cultured with PANC-1 target cells and serially diluted peptides for 24 h. Co-cultured supernatants were analyzed by ELISA for secreted IFN-γ. Each data point represents the mean concentration of IL-2/IFN-γ for each sample run in triplicate wells in (**b**), (**c**), (**e**) and (**f**), and the data are representative of $n = 3$ independent experiments. **g** The 1-2C TCR-T cells were co-cultured with PANC-1 cells stably expressing the wildtype *KRAS* genes or *KRAS* G12V, G12C, or G12D mutants. Responses of 1-2C TCR-T cells with

wildtype CFPAC-1 cells or CFPAC-1 cells stably expressing HLA-A*11:01 (**h**), or with wildtype SW-620 cells or SW-620 cells stably expressing HLA-A*11:01 (**i**).
**j–l** Reactivity of 3-2E TCR-T cells against PANC-1, CFPAC-1, and SW-620 as indicated in (**g–i**). Each dot of (**g–l**) represents one technical replicate ($n = 1$ experiment). The responses were evaluated with 1-2C or 3-2E TCR-T cells prepared with T cells from $n = 3$ separate donors run independently (Donor 1-3) from (**g**) to (**l**). The columns show means of the three technical replicates. (**m-r**) Luciferase-transduced cell lines were co-cultured with mock-T or TCR-T for 48 h at various E: T ratios. The % specific lysis of the wild type tumor cell lines (black) and over-expression tumor cell lines (blue) obtained by bioluminescence assay is plotted against multiple E:T ratios. Dots of (**m–r**) represents three technical replicates ($n = 1$ experiment), data shown are representative of responses from $n = 3$ separate donors run independently (Donor S1-S3). Source data are provided as a Source Data file.

donors were tested against PANC-1 cells loaded with these homologous peptides, and no cross-reactivity was observed for any of the homologous peptides (Supplementary Fig. S10). These results suggest a low possibility of off-target toxicity for the 1-2C and 3-2E TCRs to cross-react with homologous self-antigens.

### The anti-tumor effects of 1-2C TCR in a xenograft model

The above analyses revealed that the 1-2C TCR has higher binding affinity and immune responsiveness against tumor cells than the 3-2E TCR, and 1-2C was thus selected for further in vivo anti-tumor evaluations. To investigate the in vivo anti-tumor potency of 1-2C TCR-T cells, we subcutaneously engrafted PANC-1 cells stably expressing the KRAS-G12V mutant gene in a NOD-*Prkdc*$^{em26CdS2}$*Il2rg*$^{em26Cd22}$/NjuCrl (NCG) mouse model (Fig. 5a). The 1-2C TCR-T cells were constructed with the PBMCs from two donors, and three varied dose groups (i.e., high, medium, and low doses with $1 \times 10^7$, $1 \times 10^6$, and $1 \times 10^5$ TCR-T cells, respectively) were infused. Mock-T cells ($1 \times 10^7$ cells) without exogenous TCR transduction were used as negative controls. We found that the tumor weights in the high-dose group were significantly lower than that of mock-T cell group ($P < 0.01$), whereas the medium- and low- dose groups did not display significant differences compared to the mock-T cell treatment group. The tumor volumes in the high- and medium-dose groups were substantially lower than that in the mock-T cell group ($P < 0.01$), whereas no substantial tumor suppression efficacy was observed in the low-dose group (Fig. 5b–g).

We further investigated the anti-tumor efficacy in a SW-620 xenograft model and subcutaneously engrafted SW-620 cells stably expressing HLA-A*11:01 (SW-620-HLA-A11) in NCG mice. Anti-PD-1 antibodies are widely used as an immune modulating drug, and therefore, the mice were treated with TCR-T cells co-administrated with or without anti-PD-1 antibodies to achieve a better tumor inhibition efficacy (Fig. 5h)[35]. The upregulated expression of PD-1 in 1-2C TCR-T cells and enhanced expression of PD-L1 in SW-260 cells were observed upon co-culturing of effector and target cells (Supplementary Fig. S11). PBS and mock-T cells without exogenous TCR transduction were used as negative controls. Anti-PD-1 antibody treatment in the absence of T cells did not show any anti-tumor efficacy (Supplementary Fig. S12). The results revealed that administration of 1-2C TCR-T cells resulted in significant tumor growth inhibition compared to the mock-T cell treatment group ($P < 0.001$) (Fig. 5i–o). Apart from one mouse in the mock-T and anti-PD-1 antibody co-treatment group with an exceptional higher tumor load, the tumor volumes in this group were substantially lower than that in the mock-T cell treatment group and were comparable to that in the TCR-T treatment group. When co-administrated with anti-PD-1 antibody, the average tumor volume was substantially lower than the 1-2C TCR-T cell single treatment group ($P < 0.05$).

The mice were euthanized 26 days after tumor cell implantation, and tumor tissues were subsequently excised and weighted. We observed that tumors from the TCR-T cell treatment group were substantially smaller than that in the mock-T cell treatment group,

whereas the TCR-T cell plus PD-1 antibody treatment group showed the highest substantial tumor suppression efficacy (Fig. 5i, j). The tumor weights in the 1-2C TCR-T treatment group were significantly lower than in the mock-T treatment group ($P < 0.05$) (Fig. 5i). Of note, co-administration of 1-2C TCR-T cells and PD-1 antibodies resulted in substantially decreased tumor weight compared to either the mock-T cell or the mock-T plus PD-1 antibody groups ($P < 0.01$ for both groups) and exhibited lower trends than TCR-T cell treatment alone. Further, IFN-γ ELISPOT assays revealed that splenocytes from representative 1-2C TCR-T cell treatment mice showed specific responses against the KRAS-G12V-9 peptide, indicating the persistence of 1-2C TCR-T cells in mice receiving TCR-T cells for ~1 month (Supplementary Fig. S13). These findings indicate that 1-2C TCR-T cells mediated in vivo anti-tumor activity against solid tumors, and synergetic effects could be observed when co-administrated with PD-1 antibodies.

In the above in vivo experiments, all the mice were euthanized at experimental endpoint and survival advantage could not be observed. To further investigate whether 1-2C TCR-T also has a survival advantage, we therefore performed additional experiments in SW-620 model (Supplementary Fig. S12). For a more sensitive and objective measurement of tumor burden, SW-620-A11-luci tumor cells (SW-620 expressing HLA-A*11:01 and luciferase) were constructed and in vivo luminescence imaging were used for tumor monitoring. The monitoring of tumor burden was extended to day 74 and the results showed that 1-2C TCR-T treated group showed substantial survival advantage compared with control groups (Supplementary Fig. S12).

### Overall structure of the TCR and KRAS-G12V-9 pHLA complexes

To further illustrate the presentation mechanisms of KRAS-G12V-9 in the context of HLA-A*11:01 and the recognition mechanisms of the 1-2C and 3-2E TCRs, the structures of the 1-2C/HLA-A*11:01-KRAS-G12V-9 and 3-2E/HLA-A*11:01-KRAS-G12V-9 complexes were determined at similar resolutions of 3.3 Å and 3.5 Å, respectively (Supplementary Fig. S14, Supplementary Fig. S15, and Table 1).

The overall structures and binding footprints indicate that the 1-2C and 3-2E TCRs recognize cognate pHLA in a similar mode to conventional αβ TCRs (Fig. 6a, b, e, f)[33]. The 1-2C/HLA-A*11:01-KRAS-G12V-9 complex buries a total surface area of 1903.3 Å², and the buried surface area on Vα (973.5 Å2; 51.1%) is similar to that on Vβ (929.8 Å²; 48.9%). However, the buried surface area on Vα (1113.0 Å²; 59.5%) is substantially greater than that on Vβ (757.9 Å²; 40.5%) for the 3-2E TCR.

The distribution of interactions revealed that CDR loops from the 1-2C TCR α-chains dominate the interaction with HLA-A*11:01, whereas CDR loops from the 3-2E TCR α- and β-chains contribute similarly to the binding to HLA-A*11:01 (Fig. 6d, h; Supplementary Table S4). Overall, the KRAS-G12V-9 peptides were dominantly recognized by CDR3 loops from both the α- and β-chain of 1-2C and 3-2E TCR (Figs. 6b, f). For peptide recognition, the CDR loops of 1-2C (in gray) adopt distinct conformations from that of the 3-2E TCR (Fig. 6i). Specifically, the residues from CDR3β of the 1-2C TCR (74.2%) dominate

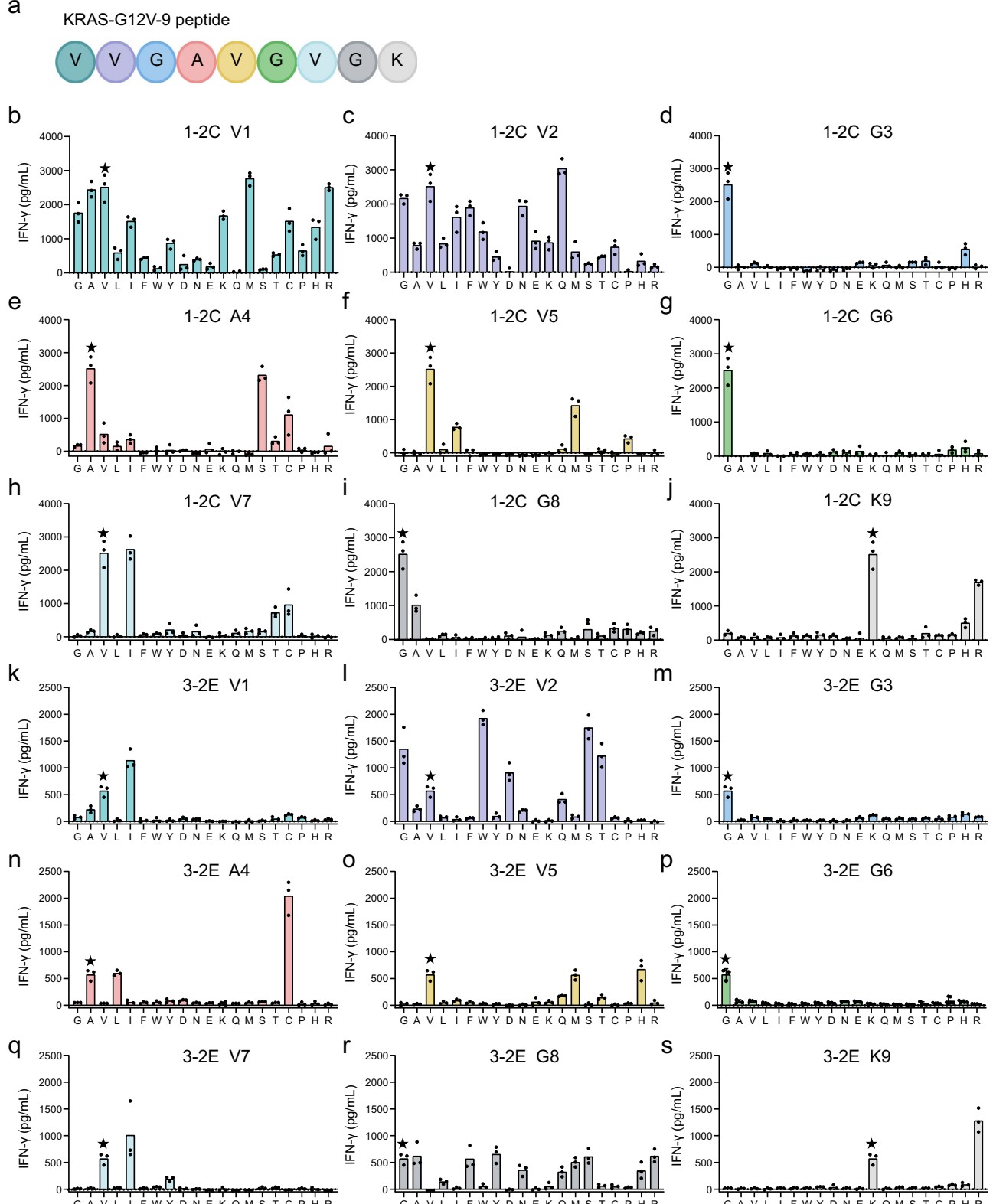

**Fig. 4 | Specificity of 1-2C and 3-2E TCR-T cells to combinatorial peptide library screening of KRAS-G12V-9. a** Amino acid positions of KRAS-G12V-9. (b-s) 9-mer combinatorial peptide library screening with PANC-1 and KRAS-G12V-9 peptide reactive 1-2C or 3-2E TCR-T cells, showing the reactive amino acid residue landscape (KRAS-G12V-9 peptide shown with a black star). Responses of the 1-2C (**b**–**j**) and 3-2E (**k**–**s**) TCR-T cells to the combinatorial peptide library were presented as indicated. Secreted IFN-γ was analyzed by ELISA with the triplicate wells of co-cultured supernatants of TCR-T cells and PANC-1 cells in the presence of the indicated peptides. The dots represent three technical replicates ($n = 1$ experiment) of the responses from one representative donor. The columns show means of the three technical replicates. Data are representative of the responses from $n = 3$ separate donors run independently (Donor S1-S3). The responses were investigated with 1-2C or 3-2E TCR-T cells prepared with T cells from Donor S1. Source data are provided as a Source Data file.

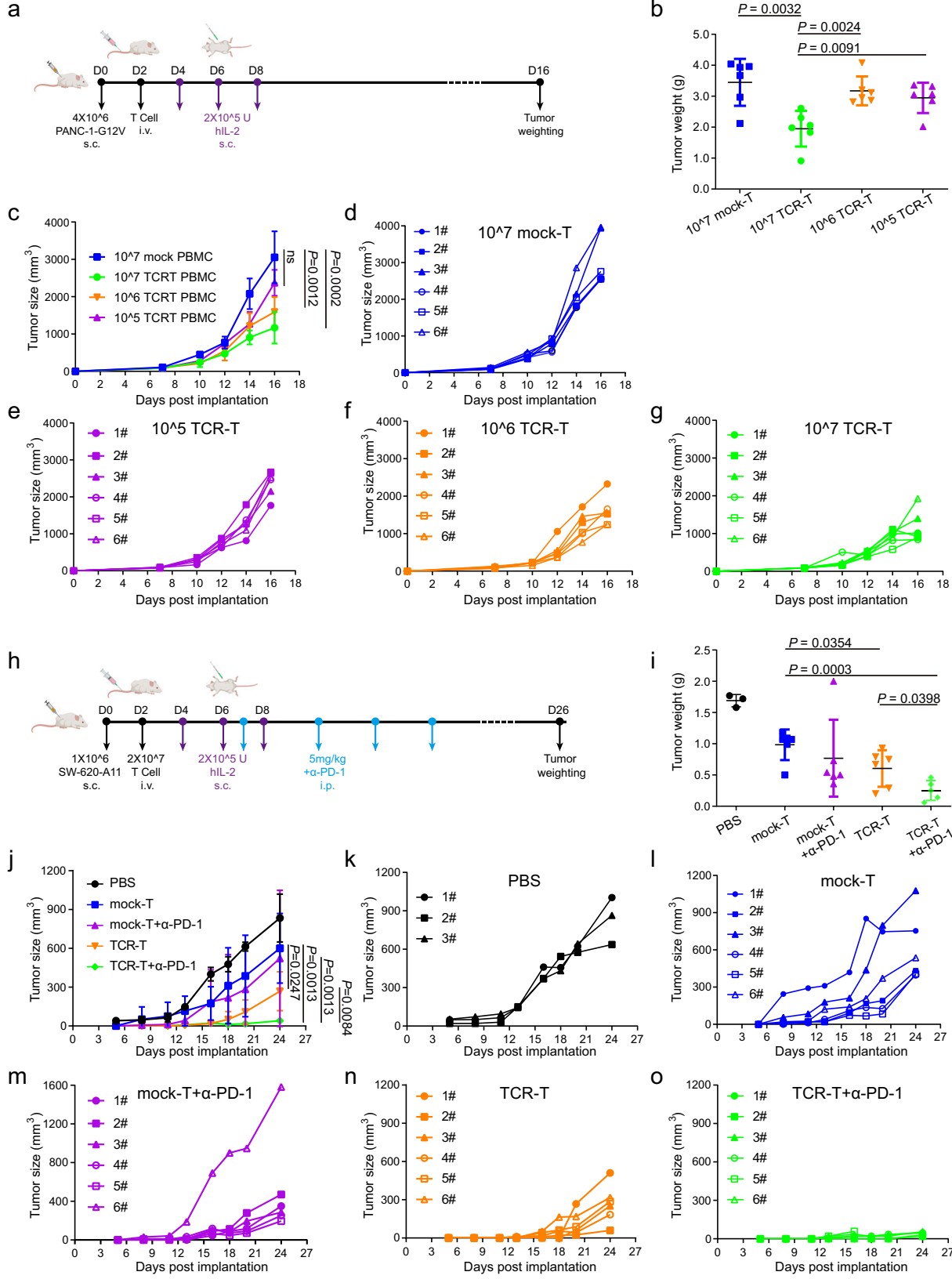

the interactions with the KRAS-G12V-9 peptide, whereas residues from both CDR3α (50%) and CDR3β (34.4%) of 3-2E make up the majority of interactions with KRAS-G12V-9 (Fig. 6k, l). These results indicate that the overall binding of 1-2C and 3-2E to the KRAS-G12V-9 pHLA complex resembles conventional αβ TCRs, though detailed distribution of the binding loops varied between these two TCRs.

## Structural basis for specific recognition of the G12V mutant by 1-2C and 3-2E

The structural basis for specific recognition of the KRAS-G12V mutant by these two TCRs was further investigated through comparative analyses. The conformations of the KRAS-G12V-9 peptide in recognition with 1-2C and 3-2E exhibited no substantial differences (Fig. 7a).

**Fig. 5 | Tumor inhibition efficacy of 1-2C TCR-T cells in a tumor-bearing mouse model. a** Schematic of the PANC-1-G12V mouse model experimental process, created with BioRender.com. NCG mice were inoculated with PANC-1 cells stably expressing KRAS-G12V ($4 \times 10^6$) (PANC-1-G12V) subcutaneously at day 0 (D0), and three dose of TCR-T cells ($1 \times 10^7$, $1 \times 10^6$, or $1 \times 10^5$) were intravenously injected on day 2 (D2). Tumor weights were monitored at the end of the experiment after sacrificing the mice. **b** The tumor weights of each tumor group from sacrificed mice at the end of the experiment are shown, $n = 6$ mice per group. **c** Tumor volumes of five groups of mice treated with $1 \times 10^7$ mock-T, $1 \times 10^7$ TCR-T, $1 \times 10^6$ TCR-T, or $1 \times 10^5$ TCR-T. **d–g** Individual follow-up of tumor sizes is presented for each experimental group with each line showing the changes of the tumor size of each mouse. $N = 6$ mice per group. **h** Schematic of the SW-620-A11 mouse model experimental process, created with BioRender.com. Four doses of PD-1 antibodies at 5 mg/kg were administrated through peritoneal injection twice a week in two mouse groups. **i** The tumor weights of each sacrificed mice from SW-620-A11 tumor groups at the end of the experiment were shown. **j** Tumor volume of five groups of mice treated with PBS, mock-T cell, mock-T cell plus anti-PD-1 antibody, 1-2C TCR-T cells, 1-2C TCR-T cells plus anti-PD-1 antibody. Mock-T cells without TCR transduction were expanded in parallel with TCR-T cells as a negative control. **k–o** Individual follow-up of tumor sizes is presented for each experimental group, with each line showing the changes of the tumor size of each mouse. The mean tumor weights or tumor volumes of each group in (**b**), (**c**), (**i**) and (**j**) were shown as black lines while the standard deviations were represented by the error bars. Statistical analyses utilized two-tailed Student's $t$ test and the $P$ values were presented as indicated, ns, $P > 0.05$. Data are shown as means ± SD. Source data are provided as a Source Data file.

The N-terminal region of the KRAS-G12V-9 peptide (V1, V2, G3 and A4) is buried in the peptide binding groove (PBG), whereas V5, G6 and V7 in the central region are exposed for TCR recognition. The N-terminal residues form multiple interactions with residues from the PBG of HLA-A*11:01, whereas K9 at the C-terminus forms multiple hydrogen bonds and acts as a typical anchor (Fig. 7b; Supplementary Table S5).

The apo structure of KRAS-G12wt-9/HLA-A*11:01 was determined for comparative analysis of specific recognition of the G12V mutant by TCRs. Comparative analysis of the conformations of wildtype and G12V mutated peptides revealed that A4 and G5 in the KRAS-G12wt-9 shift substantially upward, whereas A4 and the mutated V5 in KRAS-G12V-9 mutant peptide exhibit a downward conformation (Fig. 7c). The other regions of the peptides display no substantial differences. The V5 of KRAS-G12V-9 is buried in the D-pocket of HLA and forms hydrogen bond interactions with R114 and Q156 in the PBG, which may favor the stability of the G12V mutant pHLA complex over the wildtype peptide (Fig. 7b). Differential scanning calorimetry (DSC) analysis revealed that the thermostability of the KRAS-G12V-9/HLA-A*11:01 pHLA complex (Tm = 58.52 °C) is substantially higher than that of the KRAS-G12wt-9/HLA-A*11:01 (Tm = 52.18 °C) (Fig. 7i, j), indicating that the V5 mutation does promote peptide stability.

The conformations of the 9-mer and 10-mer peptides were further compared to illustrate the structural differences for these two epitopes. The structure of the KRAS-G12V-9/HLA-A*11:01 complex was superimposed with the KRAS-G12wt-9/HLA-A*11:01, KRAS-G12wt-10/HLA-A*11:01, and KRAS-G12D-10/HLA-A*11:01 complexes. Comparative analyses reveals that the conformation of the 9-mer peptide is distinct from that of the 10-mer peptide (Fig. 7d). The conformation of the 9-mer KRAS-G12V mutant peptide is substantially featureless with less residues exposed for TCR recognition, whereas the central region of the 10-mer peptide bulges substantially higher than the 9-mer peptide. Of note, the conformation of the N-terminal region of KRAS-G12wt-9, which showed substantial differences with the KRAS-G12V-9 peptide, is similar to that of the KRAS-G12wt-10 and KRAS-G12D-10 peptides (Fig. 7d). Binding analysis revealed that the 1-2C and 3-2E TCRs could not bind to KRAS-G12V-10/HLA-A*11:01, and A11V TCR (a previously reported KRAS-G12V-10 specific TCR) could not bind KRAS-G12V-9 (Supplementary Fig. S16). These results suggest that the 9-mer and 10-mer peptides are distinct epitopes in the context of HLA-A*11:01 and may induce varied TCR repertoires.

We further analyzed the interactions at the interfaces of KRAS-G12V-9/HLA-A*11:01 pHLA and the TCRs to investigate the molecular mechanisms underlying the specific recognition of the KRAS-G12V mutation by the TCRs. Overall, the total interactions involved in van der Waals contacts and hydrogen bonds to KRAS-G12V-9 between 1-2C and 3-2E TCR are similar to each other (Fig. 7e, f). Detailed interaction network analysis revealed that residues from CDR3β directly interact with the V5 residue in the 1-2C TCR, whereas CDR3α contributes more contacts with V5 in the 3-2E TCR (Fig. 7g, h; Supplementary Table S6). The distinct responses against V5-substituted peptide libraries between 1-2C and 3-2E TCR-T cells also indicate varied recognition profiles by these two TCRs. These results indicate that the specific recognition of the TCRs to the KRAS-G12V-9 mutant depends both on the distinct conformation from the wildtype peptide and on direct interactions with the G12V mutant residue.

## Discussion

In the present study, we identified two TCRs specific for the KRAS-G12V-9 peptide from HLA-A*11:01 transgenic mice. Binding and functional studies demonstrated that both TCRs were specific to the KRAS-G12V-9 mutant peptide, and no cross-reactivity was observed to other mutants that frequently occur in KRAS or homologous peptides in the human genome. Both 1-2C and 3-2E TCR-T cells could specifically respond to varied tumor cells with the KRAS-G12V mutation, indicating specific recognition of the TCRs to endogenously processed KRAS-G12V-9 peptide in tumor cells. The 1-2C TCR-T cells showed tumor suppression efficacy in PANC-1 and SW-620 tumor mouse models and synergetic effects were observed in combination with anti-PD-1 antibodies. Co-administration of 1-2C TCR-T cells and anti-PD-1 antibody resulted in a reduction of tumor burden, which showed more effective anti-tumor efficacy compared to specific treatment with TCR-T cell or mock-T cell/anti-PD-1 co-treatment. Considering the wide use of anti-PD-1 antibodies in clinical treatment of multiple solid tumors, future investigations of 1-2C TCR-T cells in combination with anti-PD-1 antibodies may be promising for cancer patients with the KRAS-G12V mutation.

Notably, the 1-2C public TCR was identified from all the four mice and dominated the KRAS-G12V-9-specific T cells in two mice. Public TCRs are characterized by identical TCR α and β chain sequences identified from different individuals for a specific epitope, and have been reported in immune responses to HIV epitopes, *e.g.*, Gag-KF11, and Nef-138-8[32,36]. The presence of public TCRs and broadly cross-recognition against HIV variants indicate protective roles of public T cell clones in disease control[37,38]. Multiple factors are suggested to be involved in the occurrence of public TCRs, including convergent V(D)J recombination, recombinational biases, and the conformation of the presented peptide[39,40]. We previously reported that the featureless Nef-138-8 peptide can induce public TCRs, whereas the featured Nef-138-10 peptide (an N-terminal extended overlapping peptide to Nef-138-8) can induce a highly diversified TCR repertoire[32]. The structure of the KRAS-G12V-9 peptide determined in the present study presented as a featureless epitope in the context of HLA-A*11:01, which may be a critical factor for the induction of public TCRs amongst different mice.

Previous studies show that TCRs specific for the KRAS-G12V-9 peptide do not cross-recognize with 10-mer counterparts and vice versa, as was also observed here for 1-2C and 3-2E[17]. Though the structure of the 10-mer KRAS-G12V mutant peptide in the context of HLA-A*11:01 has not been reported, the previously reported structures of 10-mer KRAS-G12wt and -G12D mutant peptides, together with the 9-mer KRAS-G12wt and -G12V mutant peptides determined in the present study, enabled us to investigate the structural differences between 9-mer and 10-mer KRAS mutant peptides in the context of

**Table 1 | Crystallographic data collection and refinement statistics**

| | 1-2C/pMHC | 3-2E/pMHC | KRAS-G12wt-9/HLA-A11 |
|---|---|---|---|
| Data collection | | | |
| Space group | $P\,1_2\,1\,1$ | $P\,6_2\,2\,2$ | $P\,2_1\,2_1\,2$ |
| Wavelength (Å) | 0.97918 | 0.97852 | 0.97853 |
| Unit cell dimensions | | | |
| a, b, c (Å) | 156.86, 156.89, 191.97 | 117.82, 117.82, 314.73 | 49.14, 67.70, 73.10 |
| α, β, γ (°) | 77.37, 78.13, 82.27 | 90.00, 90.00, 120.00 | 90.00, 100.88, 90.00 |
| Resolution (Å) | 50.00-3.35 (3.47-3.35) [a] | 50.00-3.3 (3.42-3.30) | 50.00–2.2 (2.28-2.2) |
| Unique. reflections | 249244 | 20334 | 23921 |
| $R_{merge}$ | 0.199 (0.814) | 0. 263 (1.2) | 0.242 (0.658) |
| I/σ | 6.61 (1.94) | 8.33 (1.71) | 9.50 (4.00) |
| Completeness (%) | 99.0 (99.0) | 99.7 (100.0) | 99.9 (99.8) |
| Redundancy | 3.5 (3.5) | 9.3 (9.9) | 6.7 (6.9) |
| Refinement | | | |
| $R_{work}$ / $R_{free}$ | 0.225/0.258 | 0.236/0.256 | 0.183/0.234 |
| No. atoms | | | |
| Protein | 8199 | 6543 | 3119 |
| Ligands | 0 | 0 | 0 |
| Average B-factor (Å²) | 88.58 | 74.79 | 36.46 |
| R.m.s. deviations | | | |
| Bond lengths (Å) | 0.002 | 0.005 | 0.009 |
| Bond angles (°) | 0.490 | 0.730 | 1.210 |
| Ramachandran plot | | | |
| Favored (%) | 98.59 | 96.41 | 98.94 |
| Allowed (%) | 1.41 | 3.59 | 1.06 |
| Outliers (%) | 0.00 | 0.00 | 0.00 |

[a]Values in parentheses are for highest-resolution shell.

HLA-A*11:01[22]. Binding analysis of the TCRs identified in the present study revealed that no cross-recognition of the 10-mer peptide was observed. Therefore, these findings indicate that the 9-mer and 10-mer KRAS G12 peptides are distinct epitopes in the context of HLA-A*11:01.

The structure of the 9-mer and 10-mer KRAS-G12D peptides presented by HLA-C*08:02 also displayed distinct conformations[29]. The mutated G12D residue forms a salt bridge with R156 in the HLA α2 helix and acts as an N-terminal anchor similarly in the 9-mer and 10-mer KRAS-G12D peptides, whereas the KRAS wildtype glycine at the corresponding position is not a preferred anchor for HLA-C*08:02[20]. Therefore, the lack of a primary anchor residue in the KRAS-G12wt peptide enables the induction of specific T cells for the KRAS-G12D mutant presented by HLA-C*08:02. Further, the distinct C-terminal anchor residue of the 9-mer (Ala) and 10-mer (Leu) peptides results in distinct conformation of the C-terminal portion of the peptides presented by HLA-C*08:02. This also provides direct evidence for the specific recognition of cognate TCRs and limited cross-recognition between 9-mer and 10-mer KRAS-G12D peptides in the context of HLA-C*08:02. These results suggest that 9-mer and 10-mer KRAS mutant peptides restricted by HLA-A*11:01 or HLA-C*08:02 are distinct epitopes and likely probably induce distinct TCR repertoires.

The structures determined in the present study also revealed the molecular basis for specific recognition of the 9-mer G12V mutant peptide by varied TCRs in the context of HLA-A*11:01. We showed that the peptide conformation of KRAS-G12wt-9 is different from that of the KRAS-G12D-9 peptide. The major differences between the 9-mer wildtype and G12V mutant peptides locate at the fourth and fifth

residues, whereas the other regions resemble each other. The strong interactions between V5 of the KRAS-G12V-9 mutant peptide and residues from the binding groove of HLA favors the stability of the G12V mutant pHLA complex over the wildtype peptide. Thermostability analysis revealed that KRAS-G12V-9 has a higher Tm value than wildtype or other KRAS mutant pHLAs, indicating higher stability and presentation efficiency of the KRAS-G12V-9 mutant peptide in the context of HLA-A*11:01. A possible concern of this study is that we did not obtain an apo KRAS-G12V-9 pHLA structure and thus can not eliminate the possibility that the distinct conformation of the KRAS-G12V-9 peptide originates from TCR binding induced fit. We argue that the conformation of the bulged A4 and G5 in the KRAS-G12wt-9 peptide resembles that of the 10-mer wildtype or KRAS-G12D mutant peptides, indicating the energetical privilege of this conformation. Moreover, thermostability analysis revealed that the KRAS-G12V-9 peptide has a higher Tm than the wildtype peptide, which also suggests distinct conformations for these two peptides in the context of HLA-A*11:01.

Structural analysis reveals that the binding of 1-2C and 3-2E to the KRAS-G12V-9 peptide mainly involves CDR3α and CDR3β of the TCR, which is similar to conventional αβ TCRs. The structures also revealed that residues from CDR3β of 1-2C or residues from CDR3α of 3-2E form multiple van der walls contacts with the V5 mutant residue from the KRAS-G12V-9 peptide. Therefore, we speculate that specific recognition of the TCRs for the KRAS-G12V-9 mutant peptide relies both on the distinct conformation of the mutant peptide presented by HLA-A*11:01 and on direct interaction of the CDR loops of TCRs with G12V mutant residue.

In summary, we identified two TCRs specific for the KRAS-G12V-9 peptide and engineered TCR-T cells that displayed specific responses to varied tumor cells with the KRAS-G12V mutation. The 1-2C TCR-T cells exhibited in vivo tumor suppression efficacy and improved anti-tumor potency in combination with anti-PD-1 antibodies. Structural analysis revealed that the conformation of the KRAS-G12V-9 peptide is distinct from the KRAS-G12wt peptide and the G12V mutant residue forms multiple interactions with amino acids from the TCRs, indicating the structural basis for specific recognition of KRAS-G12V mutation by TCRs. The structural basis for distinct presentation and specific recognition of the 9-mer KRAS-G12V mutant would shed light for future design of therapeutics by targeting the KRAS-G12V mutant. Our work thus provides a promising candidate with therapeutic potential against tumors carrying shared KRAS G12V mutation.

## Methods

### Cell lines, murine splenocytes, peptides

HEK-293T cells were cultured in DMEM (Gibco, cat.C11995500BT) supplemented with 10% FBS (Gibco, cat.10437-028), 100 μg/mL streptomycin and 100 IU/mL penicillin (Invitrogen, cat. 15140122). K562 and Jurkat E6 cells were grown in RPMI 1640 media supplemented with 10% FBS (Gibco, cat. 10437-028), 100 μg/mL streptomycin and 100 IU/mL penicillin (Invitrogen, cat. 15140122). Murine splenocytes were cultured in RPMI 1640 (Gibco, cat. 21875-034) media supplemented with 10% FBS (Gibco, cat. 10437-028), 100 μg/mL streptomycin and 100 IU/mL penicillin (Invitrogen, cat. 15140122), and 50 IU/mL rhIL-2 (peprotech, cat. 200-02-250UG). All cells were incubated at 37 °C in a 5% $CO_2$ environment. The cell lines were obtained from the Cell Resource Center, Peking Union Medical College.

The peptides in this study were all manufactured by the business (Genscript, Nanjing). HPLC examination revealed that the quality of the synthesized peptides was greater than 98%. Peptides were dissolved in DMSO (Amresco, cat. 0231-500 ML) and diluted in DMEM medium. Cross-reactivity of predicted epitopes with human peptides BLASTP analysis (https://blast.ncbi.nlm.nih.gov/Blast.cgi) was conducted to find KRAS-G12V-9 epitopes that match with human peptides with over 70% similarity. Peptides shared at least 6 identical

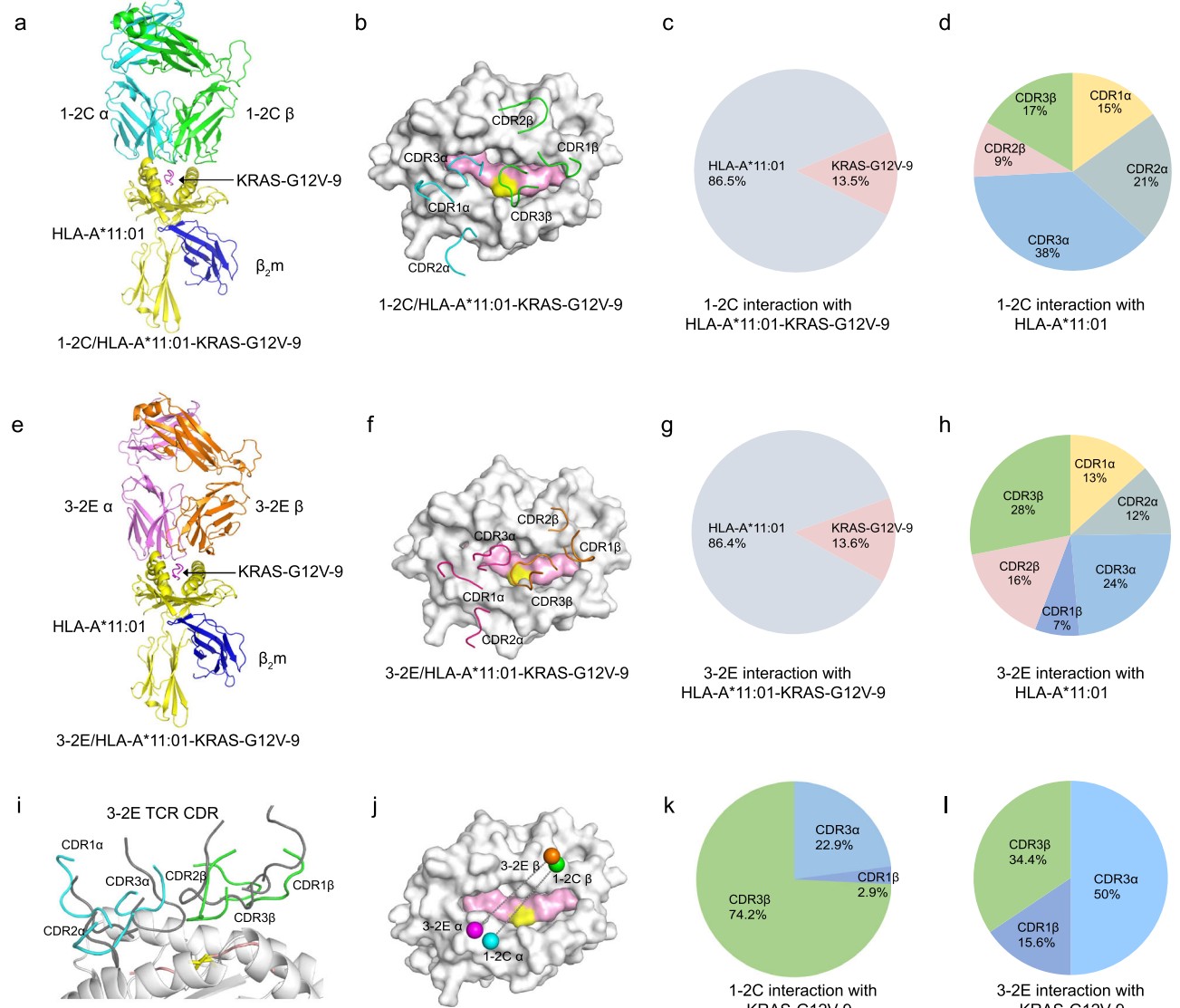

**Fig. 6 | Structure of the 1-2C and 3-2E TCRs bound to KRAS-G12V-9/HLA-A*11:01.**
**a–d** Structure of 1-2C bound to KRAS-G12V-9/HLA-A*11:01. **a** Overall structures of the 1-2C TCR in complex with the KRAS-G12V-9/HLA-A*11:01 pHLA ligand. The 1-2C TCR is shown in cyan (α chain) and green (β chain). HLA-A*11:01 is shown in pale yellow (heavy chain) and blue (β₂m), and the peptide is shown in pink (KRAS-G12V-9). **b** The footprints of 1-2C on the KRAS-G12V-9/HLA-A*11:01 pHLA complex. The three CDRαs of 1-2C are represented as ribbons in cyan, while the three CDRβs are in green. HLA-A*11:01 is depicted as a surface in gray and the KRAS-G12V-9 peptide is presented as a surface in pink. **c** The distribution (%) of HLA-A*11:01 and KRAS-

G12V-9 peptide for the recognition of 1-2C TCR. **d** The distribution (%) of the CDR loops of 1-2C for the interaction with HLA-A*11:01. **e–h** The binding of the 3-2E TCR with the KRAS-G12V-9 peptide is presented similar to that of 1-2C in (**a–d**). **i** Comparison of the CDRs of the 3-2E TCR to that of 1-2C when bound with KRAS-G12V-9/HLA-A*11:01. The CDRs of the 3-2E were colored in gray, whereas the three CDRα of 1-2C are presented in cyan and the three CDRβ in green. **j** The binding orientation of 1-2C and 3-2E TCR with KRAS-G12V-9/HLA-A*11:01 pHLA. **k, l** The distribution (%) of the CDR loops of 1-2C for the interaction with the KRAS-G12V-9 peptide.

amino acid residues with the KRAS-G12V-9 peptide were selected for specificity evaluation.

## Tetramer preparation

HLA-A*11:01-restricted tetramers with peptide KRAS-G12V-9, KRAS-G12-wt, and KRAS-G12V-10 were prepared as previously described[30,41,42]. In brief, the extracellular domain of HLA-A*11:01 (GenBank: AZL48402.1, 1-276) was changed by inserting a substrate sequence for the biotinylating enzyme BirA at the C-terminus of the α3 domain. In the presence of peptides, the modified HLA-A*11:01 and β₂-microglobulin were produced in *E. coli* (BL21, Efficom, cat. 21SEC1436) and refolded. Purified in vitro-renatured peptide/HLA-A*11:01 complexes were then biotinylated for 12 h at 4 °C with D-biotin, ATP, and the biotin protein ligase BirA (GeneCopoeia, cat. BI001). Superdex 200 10/300 GL gel filtration column (GE Healthcare) was used to further

purify the samples. After mixing with PE-streptavidin (BD, cat. 554061), the tetramers were created and kept at 4 °C in PBS containing 10 mM Tris-HCl (pH 8.0) (Novon, cat. ZZ02531), 150 mM NaCl (SCR, cat. 10019318), 0.5 mM EDTA (Amresco, cat. 0105-500 G), 0.2% BSA (Sigma-Aldrich, cat. 9048-46-8), and 0.09% NaN₃ (Sigma-Aldrich, cat. 26628-22-8).

## Flow cytometry

The following conjugated antibodies were used in this study: anti-mCD3 (17A2, BioLegend, cat. no. 100204), anti-mCD8 (53-6.7, BioLegend, cat. no. 100734), anti-hTCRα/β (IP26, BioLegend, cat. no.306723), anti-hCD8 (HIT8α, BioLegend, cat. no.300905), anti-hCD4 (OKT4, BioLegend, cat. no. 317431), anti-HLA-A2 (BB7.2, BioLegend, cat. 343303), anti-hCD3 (HIT3α, BioLegend, cat. no. 300325), anti-hIFN-γ (B27, BioLegend, cat. no. 506517). TCR-T cells were stained with

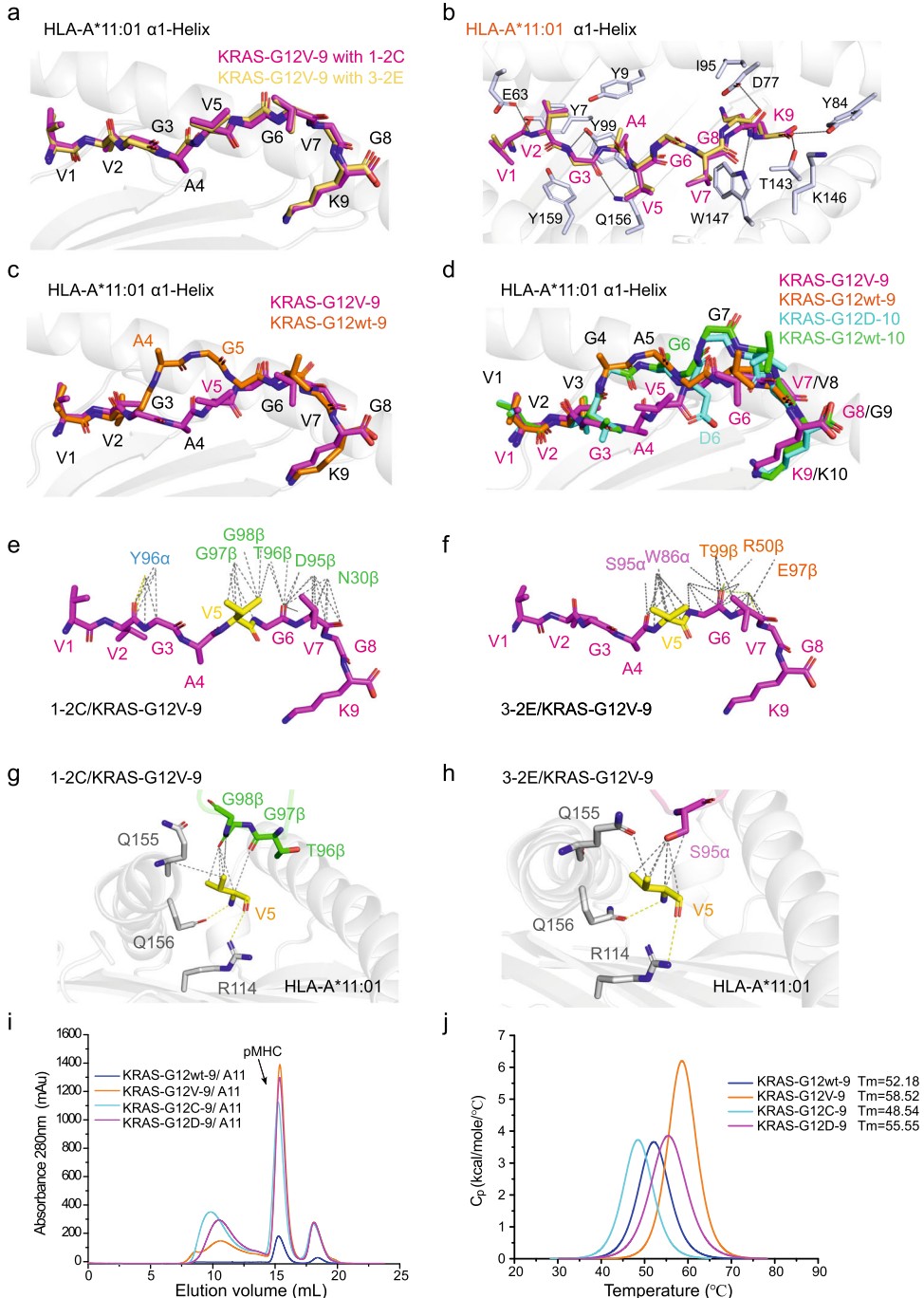

**Fig. 7 | Structural basis for the binding specificity to the KRAS-G12V mutation. a** Structure of KRAS-G12V-9 peptides in complexed with 1-2C (magenta) or 3-2E (yellow) TCR. **b** Top view of the detailed interactions of the KRAS-G12V-9 peptide with residues in HLA-A*11:01, with hydrogen bond interactions depicted as black lines. **c** Comparison of the structure of KRAS-G12wt peptide (orange) with KRAS-G12V-9 mutant peptide (magenta) presented by HLA-A*11:01. **d** Comparison of the 9-mer KRAS-G12wt (orange) and G12V mutant (magenta) peptides, and the 10-mer KRAS-G12wt (green) and G12D mutant (cyan) peptides. Detailed interactions of the 1-2C (**e**) and 3-2E (**f**) TCRs with the KRAS-G12V-9 peptide. Residues of the TCR are represented in letter-number format. Detailed interactions of the mutated V5 residue of the KRAS-G12V-9 peptide with residues from the 1-2C (**g**) or 3-2E (**h**) TCR or residues from HLA-A*11:01. Hydrogen bonds are represented by yellow dashed lines, and Van der Waals contacts are represented by black dashed lines. **i** The formation of pHLA complex of the wild type or KRAS G12 mutant (G12V, G12D and G12C) peptides with HLA-A*11:01 were evaluated with size exclusion analysis. **j** DSC evaluated the thermal stabilities of varied pHLA complex proteins loaded with the 9-mer wildtype (G12wt) or KRAS G12 mutant (G12V, G12D and G12C) peptides. Source data are provided as a Source Data file.

0.05 μg/μL tetramer per $1 \times 10^6$ cells for 30 min in FACS buffer (PBS containing 2% FBS) at room temperature, followed by other fluorescent antibodies diluted in FACS buffer at a 1:100 ratio and stained for 20 min. FACS Calibur (BD Biosciences) and FACS Aria II (BD Biosciences) were used to analyze the samples (BD Biosciences). FlowJo.V10 was used to analyze the data.

## Immunization of HLA-A*11:01 transgenic mice and single-cell sequencing of TCRs

Transgenic C57BL/6 mice expressing the human HLA-A*11:01 gene were obtained from Jackson Lab, and housed in specific pathogen free (SPF) mouse facilities in the Institute of Microbiology, Chinese Academy of Science. Anti-HLA-ABC antibody (W6/32, Biolegend, cat:

311404) staining was used to confirm the HLA expression in HLA-A*11:01 transgenic mice. For specific-genotyping, DNA was extracted from digesting tails (0.5 cm length) treated with 50 µL of Proteinase K (Thermo, cat. 25530049) at 20 mg/mL and analyzed by PCR (Forward primer: ggagacacggaatatgaaggc; Reverse primer: gtaatccttgccgtcg-tagg). All animal experiments were approved by the Committee on the Ethics of Animal Experiments of the Institute of Microbiology, Chinese Academy of Science (IMCAS) and conducted in compliance with the recommendations in the Guide for the Care and Use of Laboratory Animals of IMCAS Ethics Committee. In brief, 8 to 12-week old mice were immunized subcutaneously with 100 µg of HLA-A*11:01-restricted peptide KRAS-G12V-9 (VVGAVGVGK), mixed with CTL adjuvant (Bio-dragon, cat. KX0210044) (Mus T1-T6), or emulsified in 100 µL of incomplete Freund's adjuvant (Sigma, cat.F5506-10ML) (Mus TF1-TF6), as described before[42,43]. One week later, a booster vaccination was administered complete Freund's adjuvant (Sigma, cat. F5881-10ML) rather than incomplete Freund's adjuvant. Mice were euthanized one week following the booster vaccine, and splenocytes were collected and cultured.

Murine splenocytes were labeled with anti-mCD3, anti-mCD8, and the KRAS-G12V-9/HLA-A*11:01 tetramer one day after culture. On a FACS Aria sorter, Tetramer + /CD8 + T cells were single-cell sorted into each well of a 96-well PCR plate, and then reverse transcribed as previously reported[30]. Single T cells were subjected to an improved technique based on the technique previously described and rapid amplification of 5′ complementary DNA ends (5′ RACE) to amplify the TCR variable (V) gene sequences[44]. Following that, a long-distance polymerase chain reaction was used to synthesize double-stranded full-length cDNA. Finally, the α and β chain V region sequences of TCRs were obtained using two rounds of nested PCR.

The PCR fragments were then sequenced after examined on 1.0% agarose gel and purified. All of the TCR sequences were assessed using the IMGT/V-Quest program (http://www.imgt.org/). After sequencing and verifying, we use the template above to amplify the leader and variable regions of the TCR α or β chain. Then we generated chimeric products of "murine variable region + human constant region" using overlapping PCR. Through a series of PCRs and restriction enzyme reactions, the chimeric TCR chains were subcloned into a pCDH-EF1-MCS-T2A-Puro vector in the final format of "chimeric TCR-P2A-chimeric TCR-T2A-puro".

### Evaluation of TCR surface expression and specificity in transfected 293 T cells
One day before transfection, 6-well plates with HEK-293T cells were plated. As previously described, cells in each well were transfected transiently with 2 µg of TCR construct plasmid and 2 µg of CD3-CD8 construct plasmid[30]. The culture medium was changed into DMEM with 2% FBS after transfected 6 hr. 24 h later, cells were stained with anti-TCRα/β, anti-hCD3, and KRAS-G12V-9/HLA-A*11:01 tetramer-PE or control tetramer-PE and analyzed by flow cytometry.

### Lentiviruses production and preparation of TCR-T cells
Lentiviruses carrying chimeric TCR or CD3-CD8 were produced separately with Lenti-X cells (Takara, cat. 632180). In brief, Lenti-X cells were plated on 15 cm plates one day before transfection. Lenti-X cells in each plate were transfected with 20 µg pLP1, 13 µg pLP2, 5 µg VSVG, and 20 µg TCR construct expressing plasmid using PEI (MKbio, cat. MX2202-1G) transfection reagent. DMEM containing 2% FBS, 100 µg/mL streptomycin, 100 IU/mL penicillin, and 0.12 % Sodium butyrate (Sigma, cat. B5887-250MG) was replaced after 12 h transfected. After 48 h of transfection, the supernatants above were collected to obtain high-titer lentiviruses.

Jurkat cells were grown on a 6-well plate a day before transduction to obtain Jurkat cells expressing particular TCR. TCR-CD8-Jurkat cells were generated by treating Jurkat cells with high-titer TCR and CD3-CD8 lentiviruses, in combination with 12 µg/mL protamine (Sigma, cat. 9009-65-8). TCR-CD8-Jurkat cells were grown for two weeks under puromycin (Invivogen, cat. ant-pr-1) pressure of 2 µg/mL to improve TCR and CD8 expression. TCR-CD8-Jurkat cells were co-incubated in a 96-well plate with K562-A11 cells in a 10:1 ratio. The supernatant was analyzed for IL-2 secretion with an ELISA kit after 48 h of co-incubation and processed according to the manufacturer's protocol (Biolegend, cat.431804).

Human T cells from healthy donors were used to prepare 1-2C and 3-2E TCR-T cells. The use of the human T cells from healthy donors was reviewed and approved by the Institute of Microbiology, Chinese Academy of Sciences of Research Ethics Committee. Written informed consent was obtained from each of the donors. PBLs were isolated from peripheral blood using Ficoll-Hypaque density gradient centrifugation (Tianjin Haoyang). 6-well plates (non-treated tissue culture plate) were coated with RetroNectin (Takara, cat.T100B) for two hours two days before viral transduction. 1-2 × 10^6 total primary T cells were added to the pre-coated plates and activated per well with Human T-Activator CD3/CD28 (ThermoFisher, cat. 11131D) in T cell medium (GT-T551 media) (Clontech) supplemented with 5% (v/v) human AB serum (Gemini, cat. 100-512), 2000 IU/mL rhIL-2 and antibiotics (pen/strep). High-titer TCR-lentiviruses and protamine sulfate (12 µg/mL) were added to transfect primary T cells after activation. After 24 h, the medium was replaced and maintained with GT-T551 T cell medium. Transduced T cells were examined for TCR surface expression after 7 days and utilized for functional tests as described below.

### Target cell preparation
K562, SW-620, PANC-1, CFPAC-1 cells were engineered and used as target cells for evaluation of specific responses of the TCRs. K562, SW-620 and CFPAC-1 cells stably expressing HLA-A*11:01 were prepared through transfection with lentiviruses with HLA-A*11:01 heavy chain. The cell lines were obtained from the Cell Resource Center, Peking Union Medical College. PANC-1 cells stably expressing KRAS-G12wt, KRAS-G12V, KRAS-G12D or KRAS-G12C genes were prepared through transfection with lentiviruses with corresponding genes. Flow cytometry was used to assess the positive frequency of transduced genes in target cell lines, which was then maintained using puromycin. Puromycin selection concentration for PANC-1-G12V, CFPAC-1-HLA-A11, and SW-620-HLA-A11 was 7.5 µg/mL, 1 µg/mL and 0.5 µg/mL, respectively. For bioluminescence-based cytotoxicity assay, wildtype and HLA-A*11:01/KRAS-G12V target cell line were prepared through transfection with lentiviruses with GFP and luciferase.

### Intracellular cytokine staining (ICS)
The ICS assay was carried out exactly as previously described[30,45,46]. Briefly, TCR-transduced T cells were co-cultured in 96-well plates with target cells for 2 h. As negative and positive controls, cells grown with medium alone or PMA/ION (Dakewe Biotech Co., cat. 2030421) were utilized. For a further 6 h at 37 °C, the cells were treated with GolgiStop (BD Biosciences, cat. 555029). Anti-CD3 and anti-CD4/or anti-CD8 surface markers were used to label cells before they were fixed and permeabilized in permeabilizing buffer (BD Biosciences, cat. 51-2090KZ) and stained with anti-IFN-γ-PE (4 S.B3, Biolegend, cat. 502509). Following washes and re-suspension, samples were examined on a FACS Aria II.

### IFN-γ ELISPOT and ELISA assay
**IFN-γ ELISPOT assay.** TCR-T cell responses were detected by IFN-γ-specific enzyme-linked immunospot assay (ELISPOT) (BD Bioscience, cat. 551849) and ELISA (BD Bioscience, cat. 555142) as previously described[30,46]. In brief, 96-well ELISPOT plates were coated overnight at 4 °C with anti-human IFN-γ antibody dilutions. TCR-T cells as effector cells (1 × 10^5 cells per well) and different tumor cells as target cells (5 × 10^4 cells per well) were mixed in wells and incubated for 18 h,

the number refered in this study was the total number of T cells. PMA/ION was used as a positive control for non-specific stimulation, and cells incubated without stimulation were used as a negative control. After incubation, cells were removed and the plates were processed according to the manufacturer's instructions. The number of spots was captured and quantified using an automatic ELISPOT reader and image analysis software (Cellular Technology Limited). After peptides immunization, the peptide-specific T cells in the mouse spleen were also determined using the IFN-γ-specific ELISPOT assay (BD Bioscience, cat. 551881).

**IFN-γ-specific ELISA assay**. The co-incubation ratio between the effector cells and the target cells was 5:1. The supernatant was tested for secreted IFN-γ using a commercial ELISA kit (Biolegend, cat. 430104) after 24 h of co-incubation.

### Bioluminescence-based cytotoxicity assay

In order to establish target cells for bioluminescence-based cytotoxicity assay, a panel of cell lines were generated via lentiviral transduction including SW620-luciferase, SW620-HLA-A11-luciferase, CFPAC1-luciferase, CFPAC1-HLA-A11-luciferase, PANC-1-luciferase, and PANC-1-KRAS-G12V-luciferase. 20,000 target cells in T-cell medium were seed in 96-well-plate co-cultured with 1-2C, 3-2E TCR-T or mock-T cells at varying effector-to-target (E: T) ratios for 48 h in triplicate. Wells containing target cells with mock-T cells served as negative controls for baseline cytotoxicity. Subsequently, the supernatant was removed and the collected cells were treated with 50 μL 1 × lysis buffer (Promega, cat.E1941) on ice for 30 min. Then, 10 μL lysate supernatant were mixed with 50 μL Luciferase reagent (Promega, cat.E1501) in a LumaPlate-96 white plate (Greiner, cat. 655074), followed by measuring BLI with a luminometer (Promega GloMax® 96 Microplate Luminometer). The % specific lysis of tumor cells was calculated using the formula: % specific lysis = (luminescence of tumor cell line cocultured with mock-T cells - luminescence of tumor cell line cocultured with TCR-T cells)/ luminescence of tumor cell line cocultured with mock-T cells × 100.

### SPR assay

TCR and pMHC protein expression and purification were carried out as previously described for SPR analysis[32,47,48]. In brief, the murine TCR variable region and the ectodomains of human TCR C genes were cloned into pET21a vectors (Invitrogen) with an artificial disulfide bond. TCR and chain proteins were produced as inclusion bodies in *E.coli* (BL21-DE3) independently. An in vitro refolding procedure was used to produce soluble TCR proteins. Then the dissolved α- and β-chain inclusion bodies were injected into a refolding buffer (5 M) urea, 400 mM L-arginine HCl, 100 mM Tris [pH 8.0], 5 mM reduced glutathione (GPC, cat. AA018-250g), and 0.5 mM oxidized glutathione (GPC, cat. AA270-100g). After that, the refolding mixture was dialyzed for 24 h in 10 volumes of ultrapure water (Milli-Q system) and then against 10 volumes of exchange buffer (10 mM Tris/10 mM NaCl [pH 8.0]). The refolded samples were purified by gel filtration on a Superdex 200 10/30 GL column (GE Healthcare) after being loaded on a Source 15Q anion exchange column (GE Healthcare). Reduced (with Dithiothreitol) and non-reduced (without Dithiothreitol) SDS-PAGE analyses of purified proteins were performed.

Similar to the tetramer production described above, biotinylated pMHCs were expressed and purified using a variety of peptides, however PE-streptavidin was not added to create a tetramer. Finally, TCR and biotinylated pMHCs proteins were buffer changed into PBST (PBS + 0.05% Polysorbate 20). Binding analysis was carried out using a Biacore 8 K machine with streptavidin chips (Cytiva, cat.BR100531). 300 to 500 response units of biotinylated pMHC proteins were immobilized on the SA chip. The chip surface was then flowed with a series of two-fold diluted TCR proteins ranging from 100 M to 6.25 M. The Multi-cycle binding kinetics was analyzed with the Biacore 8 K Evaluation Software (version1.1.1.7442) using a 1:1 Langmuir binding model.

### In vivo anti-tumor activity in a xenograft tumor model

For the tumor model, highly immunodeficient female NCG mice (Gem Pharma Tech, Nanjing, Strain NO.T001475) were employed, and all animals were housed in specific pathogen-free conditions. The mice were inoculated with SW-620-HLA-A11 cells or PANC-1-KRAS-G12V cells into one of the flanks of each mouse subcutaneously. Two days after engraftment, TCR-T cells ($2 \times 10^7$ cells/mouse) was injected *i.v.* and three times of IL-2 ($2 \times 10^5$ U/mouse) were injected *s.c.* for SW-620-HLA-A11 tumor model, whereas high dose ($1 \times 10^7$ cells/mouse), medium dose ($1 \times 10^6$ cells/mouse) and low dose ($1 \times 10^5$ cells/mouse) 1-2C TCR-T cells were injected *i.v.* for PANC-1 tumor model. Mock-T cells transduced without TCR were injected as negative control. Four doses of anti-PD-1 antibodies (2 mg/kg per dose) were injected in additional TCR-T and mock-T cell groups *i.p.* twice a week in SW-620 tumor model. Growth rates were measured by length and width of tumors with calipers every twice a week. The tumor growth was monitored twice a week and the volume of the tumors was calculated by the formula: ½ length × width². The experimental endpoint was set at the earliest stage when significant differences were observed among treatment groups. The maximum limits of tumor burden include tumor volume >4000 mm³ in mice, or tumor weight >10% of body weight, or ulceration, infection or necrosis of tumor. Mice were sacrificed with a $CO_2$ chamber. The tumors were separated from each mouse and weighted. The spleen cells from representative mice were obtained and used in ELISPOT assay to test T cell responses to KRAS-G12V-9 peptide.

To investigate the survival advantage of 1-2C TCR-T cells in tumor mouse model, SW-620-A11 luciferase-expressing cells were inoculated into the right back regions of each mouse (1×10⁶ cells/mouse) subcutaneously. Tumor burden was measured using a sensitive in vivo luminescence imaging (IVIS Spectrum, PerkinElmer) method, and total flux was computed using the provided software (Living Image, Perkin Elmer), which measures the brightness through typical circular regions of interest (ROIs). According to their initial tumor burden, mice were ranked and then randomly assigned to PBS, α-PD-1 alone, mock-T cell, or 1-2C TCR-T cell treatment groups. The tumor burden of the mice was monitored over time using in vivo luminescence imaging. The mouse with a tumor flux intensity above $2 \times 10^{10}$ was recognized as overburden and euthanized subsequently, whereas the rest mice were left for further monitoring. The experimental endpoint was set at the earliest stage when substantial survival advantage was observed for TCR-T treatment group.

Animal care was carried out in accordance with the guidelines of Animal Care and Use Committee of Institute of Microbiology, Chinese Academy of Sciences.

### Differential scanning calorimetry for protein thermostabilities studies

TA Instruments Nano DSC was used for testing the thermostabilities of HLA-A*11:01 with KRAS G12 mutant peptides. The isolated pMHC complexes were concentrated to 1 mg/mL in a buffer solution of 20 mM Tris-HCl (pH 8.0) and 50 mM NaCl. The TA Nano DSC calorimetry cell has a volume of 300 μL, and the needed sample volume was 600 μL. The buffer solution was initially put into both the reference and sample cells, and the background scanning was performed from 20 to 90 °C at a rate of 1 °C/min. The cleaning procedure is critical for producing believable and accurate differential scanning calorimetry (DSC) data. The endothermic or exothermic enthalpy was measured from 20 to 90 °C at the same rate of 1 °C/min. In most cases, the graphic shows a single endothermic peak. The peaks can be integrated to directly yield values of enthalpy calorimetric pair process ($\Delta H_{cal}$) and melting temperature ($T_m$) after calibration of the instrumental

baseline, transition baseline, and normalization to concentration[49]. The data was analyzed using TA Instruments Nano Analyze, which was paired with the instrument. OriginPro9.1 software was used to export and plot the fitting data.

**Crystal screening, data collection and structural determination**
The complex of TCR and KRAS-G12V-9/HLA-A*11:01 were produced by co-incubating TCR and pHLA proteins at a molar ratio of 1:1 and then purified using gel filtration. The TCR/pHLA complex proteins were then purified to a concentration of 10 mg/mL in preparation for crystallization. Diffraction quality crystals of the complex of TCR/pHLA was obtained by sitting drop vapor diffusion at 18 °C by mixing 1 μL of protein with 1 μL of reservoir solution. Crystals of 1-2C TCR/pHLA complex grew in 1.0 M Ammonium citrate tribasic pH 7.0, 0.1 M BIS-TRIS propane pH 7.0, and crystals of 3-2E TCR/pHLA complex grew in 0.1 M HEPES pH 7.5, 10% w/v PEG 8000/ 8% v/v Ethylene glycol. For the purpose of protecting crystals, they were stored in the anti-freezing solution (the mixture of 2.5 μL crystallization buffer and 1 μL 20% (v/v) glycerol) before flash-cooling in liquid nitrogen. Diffraction data were collected at Shanghai Synchrotron Radiation Facility (SSRF) BL19U. The HKL2000 program was used to process all of the datasets[50]. The structures of 1-2C and 3-2E TCR/pHLA complexes were determined by the molecular replacement method using the molecular replacement method using Phaser (CCP4 suite) with previously reported TCR/pHLA structure (PDB: 6UON, 6DFV, 6OVN, 7Z50) as the search models[51]. Coot and Phenix were used to finish and improve the atomic models[52,53]. The stereochemical qualities of the final model were assessed with MolProbity[54]. Pymol (http://www.pymol.org) was used to generate all structural diagrams. The coordinates and structure factor for the structure described here have been deposited in the Protein Data Bank under the PDB code (8I5C, 8I5D and 8I5E).

**Reporting summary**
Further information on research design is available in the Nature Portfolio Reporting Summary linked to this article.

# Data availability
Crystal structures reported in this study have been deposited in the Protein Data Bank under the PDB code 8I5C, 8I5D and 8I5E. The remaining data are available within the Article, Supplementary Information or Source Data file. Source data are provided with this paper.

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

## Acknowledgements
This work was supported by the National Key Research and Development Program of China (Grant No. 2022YFC2302900 and 2021YFC2301400 to S.T.), National Natural Science Foundation of China (NSFC, 92169208 and 32222031 to S.T.), Chinese Academy of Sciences (YSBR-083 to S.T.). We thank the staff of BL19U beamline at the Shanghai Synchrotron Radiation Facility for assistance with data collection. We thank Zheng Fan, and Wei Zhang from the Institute of Microbiology, CAS, for their technical support in the SPR assay. We also thank Ting Li from the Institute of Genetics and Developmental Biology, CAS, for her technical support in the flow cytometry analysis.

## Author contributions
S.T., D.L. and G.F.G. designed experiments, analyzed the data and supervised the project. D.L., Yuan Chen, M.J., W.S., Y.L., X.Z. and J.W. performed experiments. S.T., Yan Chai, D.L., K.M. and J.Q. solved the structure and analyzed the data. J.W., Yu Chen, H.L., W.J. and C.W.Z analyzed and discussed the data. S.T., D.L. and G.F.G. wrote the manuscript.

## Competing interests
The authors declare no competing interests.
