## [Peer Review File · Nature Communications]

KRAS G12V neoantigen specific T cell receptor for adoptive T cell therapy against tumorsREVIEWER COMMENTS

Reviewer #1 - Structural TCR, X-ray crystallography - (Remarks to the Author):

In this manuscript, Lu et al. describe the isolation and characterization of TCRs specific for KRAS-G12V neoepitope presented by HLA-A*11:01. These TCRs were derived from HLA-A*11:01 transgenic mice immunized with KRAS-G12V 9-mer peptide. The authors used binding and functional assays to demonstrate that two TCRs (1-2C and 3-2E) from immunized mice recognize the KRAS-G12V mutant but not wild-type KRAS. Moreover, TCR 1-2C showed tumor inhibition in a xenograft mouse tumor model. Crystal structure of TCRs 1-2C and 3-E2 in complex with KRAS-G12V/HLA-A*11:01 revealed the basis for specific recognition of the KRAS neoepitope. This study adds to previous work from other groups on TCRs specific for other HLA-restricted KRAS neoepitopes: KRAS-G12D/HLA-C*08:02 and KRAS-G12D/HLA-A*11:01. It is therefore a useful contribution to on-going efforts to develop adoptive T cell therapies against tumors expressing KRAS driver mutations. The functional assays and structure determinations are technically well done and the results support the authors' conclusions.

Points to address:

1. The TCRs in this study contain mouse, rather than human, V regions. In this respect, they differ from other reported TCRs specific for KRAS neoepitopes that were isolated from cancer patients and are therefore fully human. This is a potential concern for adoptive T cell therapy because mouse, unlike human, TCRs would not have undergone negative selection in a human thymus to remove autoreactive TCR. This important distinction should be made clear by the authors on p.12 in the section on off-target binding to homologous self-peptides in the human genome.
2. The Discussion repeats many of the same points made in the Results, especially regarding structural details of the TCR/KRAS-G12V/HLA-A*11:01 complexes. It can easily be cut in half.

Reviewer #2 - Tumour-specific T cells, KRAS - (Remarks to the Author):

In this paper, Lu et al. describes two novel TCRs directed at KRAS G12V/HLA-A*11:01. These TCRs were isolated from HLA-A*11:01 transgenic mice immunized with a 9-mer peptide corresponding to AA sequence VVGAVGVGK of mutated KRAS. The investigators demonstrate TCR recognition of p/HLA complexes by surface plasmon resonance binding and flow cytometry assays. The functionality of these TCRs is tested upon expression as chimeric receptors in either Jurkat or primary CD8+ T cells. Upon challenge with peptide or G12V+ cancer cell lines matched for HLA-A*11:01 expression, TCR-expressing T cells are activated and produced IL-2 and IFN-g. Adoptive transfer of the TCR-expressing (2×10^7) T cells results in decreased tumor growth up to day 24. The investigators also provide 3D information (3.3-3.5 Å resolution) on the interaction of these TCRs with p/HLA complexes and propose a structural basis for antigen binding specificity of these TCRs.

General Comments:

The 2 TCRs described in this paper complement a previous study by Wang et al. (PMCID: PMC4775432) describing 2 TCRs also isolated from A*11 transgenic mice. Of significance, there is sharing of TRVB family (BV13-2*01) between TCR 1-2C described by Lu and one of the TCRs previously described by Wang.

Both 9-mer and 10-mer G12V peptides are presented by HLA-A*11:01 (PMCID:PMC9017224). What was the rationale for only immunizing mice with 9-mer? Wang et al reported on TCRs recognizing 9-mer and 9-mer/10-mer G12V peptides.

The 3D data described in this paper is novel. Unfortunately, the investigators chose to compare a G12V 9-mer with a G12D 10-mer. Side chains for these 2 AA are different in size and polarity. The significance of this comparison and how it aids in understanding the structural basis of G12V 9-mer vs 10-mer peptide binding and TCR recognition is missing.

Specific Comments:

1-Figure 3 is informative, but results represent activation of TCR-expressing cells. The recognition and killing assays reported on Figure S5 are more significant and should be expanded to other cancer cell lines.

2- Off-tumor recognition by TCR is a serious safety concern. The analysis presented on

Figure 4 is incomplete. Selection of peptides based on AA sequence similarity is insufficient as TCR cross-reactivity can be observed to AA sequences lacking sequence similarity (PMCID: PMC3743463). Studies using a combinatorial peptide library seems more appropriate given that these TCRs were isolated from mice.

3- In Figure 5, what is the effect of anti-PD1 alone? This condition appears to be missing. How was the concentration of 2×10^7 T cell dose chosen? Can experiments be expanded beyond 24 days, does treatment confer survival advantage?

In general details as to the number of times experiments were performed are missing. Similarly, statistical analysis is missing in most data.

Reviewer #3 - Cancer immunotherapy, ACT - (Remarks to the Author):

The study by Lu and colleagues describes the generation of mutant KRAS-specific TCRs restricted to the HLA-A*11:01 in HLA-transgenic mice. KRAS G12V-specific T cells were labelled with tetramers and sorted from mouse splenocytes. The TCRs were isolated and sequenced before cloning into lentiviral expression vectors. The TCRs were evaluated in both CD8+ Jurkat cells and primary T cells.

Two of the TCRs that were well expressed and specific were selected for further study and tested against four cell lines transduced with HLA-A*11:01 and one also with different KRAS mutations.

The authors have very thoroughly evaluated the TCRs, studying functionality of transduced T cells in vitro and in xenograft models, as well as looking at TCR affinity and pMHC stability and the structure of TCR and pMHC. The study is very well conducted and the conclusions supported by the data. However, some of the data and methods require some clarification.

1. In Figure 1 the tetramer staining of TCR-expressing HEK-293T cells. The transduction efficiency was variable for the different TCRs and two TCRs were chosen for further study. In Figure 2 the HEK-293T cells expressing the two TCRs are assessed by tetramer staining for the 7 KRAS mutated 9-mer peptides and the corresponding wt peptide. Do all peptides bind the HLA-A*11:01 molecules equally well?

2. In Figure S3 the intracellular cytokine production of TCR transduced T cells is assessed in response to PANC-1 peptide loaded target cells. Primarily CD8 T cells respond, but the

response rate is very low (2.85% IFN-g+ for I-2C TCR and 1.9% for 3-2E). What was % of TCR-expression in the evaluated T cells? Is there additional evidence for the CD8 dependency of the TCRs? The transduction efficiency also seemed to be quite low for the two TCRs in certain donors (Figure S4).

3. PANC-1 cells were modified to express different KRAS mutations. Their expression was assessed by flow cytometry (Figure S4), however the results are not shown and the details of the flow cytometry not outlined. Which anti-HA antibody was used, were the cells sorted as the expression of the different constructs was variable in the cell line?

4. In Figure 3 IFN-g ELISPOT assays were used to test both TCR-transduced T cells and assess the frequency of peptide-specific T cells in the spleen of immunized mice (Fig. S6). The ELISPOT assay for human T cells is fairly well described with reference to previous papers, however, it is not explained why the spot counts are expressed as number of spots per 10e6 cells as 10e5 T cells were plated per well.

The y-axis is labelled with TCR+ cells, does this mean the percentage of TCR expression was taken into account for the E:T ratios or does this refer to total number of T cells?

Was the expression level of the mutated KRAS the same for the PANC-1 cells (see point 3) in Figure 3 g and j)?

The ELISAs show a clear dose-dependent response to the KRAS G12V peptide, with a higher affinity for the 1-2C TCR consistent with the SPR data presented in Figure 2.

5. In Figure 4 where cross-reactivity to peptides with similar sequences was tested, donor 1, 2 3 are also used. Were these the same as the donors shown in Figure 3 and S4? Which peptide concentration was used in the assay presented?

Were any other methods or additional cytokines (TNF-a, IL-2) or degranulation markers (CD107a) used to verify cross-reactivity? How about other cell lines or healthy tissues?

6. Very few details about the ELISPOT used to test the murine T cells after immunization are provided. This cannot be the same kit, please provide details of the kit and cell numbers used (only E:T ratio is provided. Were conditions tested in triplicate? Only one well per condition is shown. The positive control (PMA/ionomycin) did not work well in all cases, please comment.

7. In vivo: NCG mice are used with a human xenograft (SW620) and human T cells presumable. However, the heading in the methods says syngeneic tumor model. Please correct.

The in vivo results in Figure 5 show potent anti-tumour reactivity of the I-2C T cells. How many times were T cells tested in NCG mice?

Anti-PD-1 treatment was also included in the in vivo experiment and showed some synergy with the T cell treatment, even in the group treated with mock T cells + anti-PD-1. Apart from one mouse in this group with a higher tumour load, the anti-PD-1 treatment was as efficacious as TCR transduced T cells without anti-PD-1. This should be discussed. Was PD-1 expression detected on the T cells? Did the SW620 cells express PD-L1?

ELISPOT assays were carried out to assess the functional response of splenocytes in mice sacrificed on day 26. Was the TCR expression of the splenic T cells also assessed?

Why were T cells from the TCR/anti-PD-1 treated group not tested in IFN- γ ELISPOT?

Why were mice sacrificed on day 26? Was survival assessed in any of these mice according to humane endpoints?

8. The structural data in Figure 6 are interesting and the increased thermostability (Figure S9) of the KRAS-G12V -9/HLA-A*11:01 complex compared to the complex with wt KRAS peptide would be important for KRAS-specific TCR-based therapy.

9. The alanine scan performed in Figure 7 indicated that virtually all peptide positions were important for a functional T cell response. The alanine replacements could also disrupt the binding of the peptide to HLA-A*11:01, was this assessed in any way?

10. The English language needs editing in parts of the paper. Some sentences need grammatical improvement and some wording needs changing (e.g. mice were sacrificed rather than slaughtered).

The abstract should be rewritten and the conclusions could be bolder regarding future prospects of TCR-based therapy.

REVIEWER COMMENTS

Reviewer #1 - Structural TCR, X-ray crystallography - (Remarks to the Author):

In this manuscript, Lu *et al.* describe the isolation and characterization of TCRs specific for KRAS-G12V neoepitope presented by HLA-A*11:01. These TCRs were derived from HLA-A*11:01 transgenic mice immunized with KRAS-G12V 9-mer peptide. The authors used binding and functional assays to demonstrate that two TCRs (1-2C and 3-2E) from immunized mice recognize the KRAS-G12V mutant but not wild-type KRAS. Moreover, TCR 1-2C showed tumor inhibition in a xenograft mouse tumor model. Crystal structure of TCRs 1-2C and 3-2E in complex with KRAS-G12V/HLA-A*11:01 revealed the basis for specific recognition of the KRAS neoepitope. This study adds to previous work from other groups on TCRs specific for other HLA-restricted KRAS neoepitopes: KRAS-G12D/HLA-C*08:02 and KRAS-G12D/HLA-A*11:01. It is therefore a useful contribution to on-going efforts to develop adoptive T cell therapies against tumors expressing KRAS driver mutations. The functional assays and structure determinations are technically well done and the results support the authors' conclusions.

Points to address:

1. The TCRs in this study contain mouse, rather than human, V regions. In this respect, they differ from other reported TCRs specific for KRAS neoepitopes that were isolated from cancer patients and are therefore fully human. This is a potential concern for adoptive T cell therapy because mouse, unlike human, TCRs would not have undergone negative selection in a human thymus to remove autoreactive TCR. This important distinction should be made clear by the authors on p.12 in the section on off-target binding to homologous self-peptides in the human genome.

Reply: We agree that the concerns of off-target toxicity do exist. As is also suggested by Reviewer 2, we therefore further performed tests against combinatorial peptide libraries to address this concern. A combinatorial peptide library was synthesized with each of the residues substituted with 20 amino acids.

The responses of 1-2C and 3-2E TCR-T cells constructed from three donors were tested with IFN- γ -ELISA assay. The responses of the 1-2C and 3-2E TCR-T cells showed similar profiles among the two donors and the data from one donor were presented in revised Figure 4, whereas the data obtained from the other donor were presented as Figure S8.

The results showed that the responses against G3 and G6 were highly specific that no substantial responses could be observed when substituted with other amino acids. Many of the substitutions at V1 and V2 were tolerated for 1-2C TCR-T cells, as is also observed for the substitutions at V2 and G8 for 3-2E TCR-T cells. Cross reactivity at the C-terminal anchoring K9 could only be observed against substitution with Arg for both of the TCRs, which was in line with the binding motif of HLA-A*11:01 restricted peptides. For G12V mutant position substitutions, substantially decreased responses for 1-2C TCR-T cells could be observed against substitutions with Ile, Met and Pro, whereas no responses could be observed with other substitutions. Responses of 3-2E TCR-T cells could only be observed for Met and His at G12V mutant position. Overall, the responsive profiles at V2, G3, A4, V5, G6, V7 and K9 were similar for both 1-2C and 3-2E, whereas responses at V1 and G8 substantially varied for these two TCRs. We further asked whether homologous peptides exist in human genome for these site-substituted peptides which showed cross-reactivity. Through blasting analysis, none of these peptides exists in human genome.

Therefore, we believe these results would benefit for our understanding of the recognition of these two TCRs and for further investigation of potential off-target cross-reactivity. We have also added description on p.12 to make this clear as “The TCRs identified in the present study were from HLA-A*11:01 transgenic mouse and the concerns referred to off-target toxicity to self-antigens in human cannot be fully excluded.”

2. The Discussion repeats many of the same points made in the Results, especially regarding structural details of the TCR/KRAS-G12V/HLA-A*11:01 complexes. It can

easily be cut in half.

Reply: We agree with this suggestion and have removed many of the structural descriptions in discussion section.

Reviewer #2 - Tumour-specific T cells, KRAS - (Remarks to the Author):

In this paper, Lu et al. describes two novel TCRs directed at KRAS G12V/HLA-A*11:01. These TCRs were isolated from HLA-A*11:01 transgenic mice immunized with a 9-mer peptide corresponding to AA sequence VVGAVGVGK of mutated KRAS. The investigators demonstrate TCR recognition of p/HLA complexes by surface plasmon resonance binding and flow cytometry assays. The functionality of these TCRs is tested upon expression as chimeric receptors in either Jurkat or primary CD8+ T cells. Upon challenge with peptide or G12V+ cancer cell lines matched for HLA-A*11:01 expression, TCR-expressing T cells are activated and produced IL-2 and IFN- γ . Adoptive transfer of the TCR-expressing (2×10^7) T cells results in decreased tumor growth up to day 24. The investigators also provide 3D information (3.3-3.5 Å resolution) on the interaction of these TCRs with p/HLA complexes and propose a structural basis for antigen binding specificity of these TCRs.

General Comments:

The 2 TCRs described in this paper complement a previous study by Wang et al. (PMCID: PMC4775432) describing 2 TCRs also isolated from A*11 transgenic mice. Of significance, there is sharing of TRVB family (BV13-2*01) between TCR 1-2C described by Lu and one of the TCRs previously described by Wang.

Reply: We have also noticed this study and cited accordingly. We described in the manuscript that the 1-2C TCR (TRAV7-2*02, TRBV13-2*01) is a public TCR that presents in each of the four mice and dominates KRAS-G12V-9 specific TCRs in two mice. It seems that these TCRs were from germline coded TCRs with less random insertions in CDR3 region. The structures of both 1-2C and 3-2E with KRAS-G12V/HLA-A*11:01 were determined in this study, indicating distinct

binding modes of these two TCRs. One of the TCRs identified by Wang et al is TRBV13-1*02 and specific to 10-mer G12V peptide, and this TCR is distinct from the two TCRs specific to 9-mer G12V peptide identified in this study. Therefore, the KRAS-G12V is highly immunogenic in HLA-A*11:01 transgenic mice with pre-existing germline encoded public TCRs, though specific TCR repertoires in human need to be further investigated.

Both 9-mer and 10-mer G12V peptides are presented by HLA-A*11:01 (PMCID:PMC9017224). What was the rationale for only immunizing mice with 9-mer? Wang et al reported on TCRs recognizing 9-mer and 9-mer/10-mer G12V peptides.

Reply: We have also tried to identify 10-mer G12V peptide specific TCRs and immunized mice with both 9-mer and 10-mer G12V peptides, but failed to obtain immune positive mice specific to 10-mer G12V peptide.

We have also noticed that Wang et al reported one TCR (TRBV4*01) that could cross-recognize 9-mer/10-mer G12V peptides, and the other TCR (TRBV13-1*02) could specifically recognize 10-mer G12V peptide. However, we found that the structures of 9-mer and 10-mer G12V peptides in the context of HLA-A*11:01 were distinct from each other and the binding analysis also revealed that both of the two TCRs identified in this study could not bind to 10-mer G12V peptides, suggesting distinct epitopes of these two overlapped peptides. Though possible processing of 10-mer peptide to 9-mer peptide may occur in antigen presenting cells, cross responses of the TCR-T cells tested in the present study did not support this hypothesis. Therefore, we propose that the cross-recognition mechanisms of the TCR (TRBV4*01) identified by Wang et al should be further investigated.

The 3D data described in this paper is novel. Unfortunately, the investigators chose to compare a G12V 9-mer with a G12D 10-mer. Side chains for these 2 AA are different in size and polarity. The significance of this comparison and how it aids in understanding the structural basis of G12V 9-mer vs 10-mer peptide binding and TCR recognition is missing.

Reply: We agree that this is a potential concern for the conclusion of the distinct presenting structures of G12V 9-mer and 10-mer peptides. We have also made this clear in the Discussion section. As overlapped peptides, the anchoring residues were critical for the determination of the peptide conformation exposed in central region for TCR recognition. The structural comparison of 9-mer G12V and 10-mer G12D shows that the two N-terminal residues (V1 and V2) and the three C-terminal residues (V7, G8 and K9 in 9-mer peptide, or V8, G9 and K10 in 10-mer peptide) of these two peptides showed similar conformations. This indicates that the anchoring regions were similar for these two overlapped peptides, which results in the different conformation of the central region of 10-mer peptide with one additional amino acid. Together with specific binding of the two TCRs to 9-mer G12V peptide, we speculate that the conformation of 10-mer G12V peptide should be differ from that of 9-mer peptide, and the differences may also reside in the central region of the peptide.

Specific Comments:

1-Figure 3 is informative, but results represent activation of TCR-expressing cells. The recognition and killing assays reported on Figure S5 are more significant and should be expanded to other cancer cell lines.

Reply: We have performed cytotoxicity assay with additional PANC1 and CFPAC-1 target cells and have added these results in revised Figure 3. 1-2C and 3-2E TCR-T cells were constructed from three donors and co-cultured with target cell or control. The cytotoxicity assay was performed with TCR-T cells co-cultured with luciferase-expressing target cells and the target cells were lysed to measure the residual-cell-associated luciferase. Tumor lysis was quantified by analysis of residual luciferase expression in tumor cells.

The results showed that both 1-2C and 3-2E TCR-T cells exhibited substantial tumor cell lysis to HLA-A*11:01 positive PANC-1 cells stably expressing G12V mutant KRAS gene, or CFPAC-1 and SW620 cells that endogenously expressing KRAS-G12V mutant and transduced with HLA-A*11:01. The dose dependent

responses with varied E:T ratio ranged from 8:1 to 1:2 was presented in revised Figure 3.

Additionally, we also performed *in vivo* tumor suppression experiment in PANC-1 tumor cell bearing NCG mouse model to expand the anti-tumor efficacy of 1-2C TCR-T cells. 1-2C TCR-T cells were constructed with the PBMCs from two donors and three varied dose groups, high, media and low doses with 1×10^7 , 1×10^6 and 1×10^5 TCR-T cells, respectively, were infused. The results showed that the substantial anti-tumor effects could be observed in high and media dose groups, whereas no substantial tumor suppression efficacy could be observed in low dose group. Together with the cytotoxicity analysis and *in vivo* anti-tumor effects, these results indicated that 1-2C TCR was highly competent with anti-tumor efficacy.

2- Off-tumor recognition by TCR is a serious safety concern. The analysis presented on Figure 4 is incomplete. Selection of peptides based on AA sequence similarity is insufficient as TCR cross-reactivity can be observed to AA sequences lacking sequence similarity (PMCID: PMC3743463). Studies using a combinatorial peptide library seems more appropriate given that these TCRs were isolated from mice.

Reply: We agree that the concerns of off-target toxicity do exist. As is also replied to Reviewer 1, we performed tests against a combinatorial peptide library to address this concern. A combinatorial peptide library was synthesized with each of the residues substituted with 20 amino acids. The responses of 1-2C and 3-2E TCR-T cells constructed from three donors were tested with IFN- γ -ELISA assay. The responses of the 1-2C and 3-2E TCR-T cells showed similar profiles among the two donors and the data from one donor were presented in revised Figure 4, whereas the data obtained from the other donor were presented as Figure S8.

The results showed that the responses against G3 and G6 were highly specific that no substantial responses could be observed when substituted with other amino acids. Many of the substitutions at V1 and V2 were tolerated for 1-2C TCR-T cells, as is also observed for the substitutions at V2 and G8 for 3-2E TCR-T cells. Cross

reactivity at the C-terminal anchoring K9 could only be observed against substitution with Arg for both of the TCRs, which was in line with the binding motif of HLA-A*11:01 restricted peptides. For G12V mutant position substitutions, substantially decreased responses for 1-2C TCR-T cells could be observed against substitutions with Ile, Met and Pro, whereas no responses could be observed with other substitutions. Responses of 3-2E TCR-T cells could only be observed for Met and His at G12V mutant position. Overall, the responsive profiles at V2, G3, A4, V5, G6, V7 and K9 were similar for both 1-2C and 3-2E, whereas responses at V1 and G8 substantially varied for these two TCRs. We further asked whether homologous peptides exist in human genome for these site-substituted peptides which showed cross-reactivity. Through blasting analysis, none of these peptides exists in human genome.

Therefore, we believe these results would benefit for our understanding of the recognition of these two TCRs and for further investigation of potential off-target cross-reactivity. We have also added description on p.12 to make this clear as “The TCRs identified in the present study were from HLA-A*11:01 transgenic mouse and the concerns referred to off-target toxicity to self-antigens in human cannot be fully excluded.”

3- In Figure 5, what is the effect of anti-PD1 alone? This condition appears to be missing. How was the concentration of 2×10^7 T cell dose chosen? Can experiments be expanded beyond 24 days, does treatment confer survival advantage?

Reply:

1) We have performed additional experiments to illustrate the treatment effect of anti-PD-1 alone and the results has been presented in supplementary Figure S11. The anti-tumor effects of anti-PD-1 antibodies primarily rely on the existence of T cells, though tumor intrinsically expressed PD-1 may also have influences to anti-PD-1 therapy. The added results showed that anti-PD-1 alone did not show substantial anti-tumor efficacy in comparison with anti-PD-1 and mock T cell co-treatment group in mouse model.

- 2) We performed *in vivo* tumor suppression experiment in PANC-1 tumor bearing NCG mouse model before the investigation in SW620 tumor model. 1-2C TCR-T cells were constructed with the PBMCs from two donors and three varied dose groups, high, media and low doses with 1×10^7 , 1×10^6 and 1×10^5 TCR-T cells, respectively, were infused. The results showed that substantial anti-tumor efficacy could be observed for high dose group and media dose group, whereas no substantial tumor suppression efficacy could be observed in low dose group. These results were added in revised Figure 5. The tumor volumes in high dose group were not completely suppressed. Therefore, we chose 2×10^7 dose in the SW620 tumor model.
- 3) We did not follow the experiment beyond 24 days. We tried to analyze the T cell responses and tumor weight in all the groups, and therefore all the mice were sacrificed for comparative analysis. Additional experiment has been performed to extend the follow-up duration has been presented in supplementary Figure S11. Notably, on day 74, the group infused with 1-2C TCR-T exhibited a sustained and statistically significant suppression of tumor growth. It is noteworthy that two mice in the TCR-T treatment group displayed complete tumor clearance. During the procedure of this experiment, no mice experienced natural mortality. Instead, euthanasia was administered upon the attainment of a solid tumor size of 1.2 cm and a fluorescence value equal to or exceeding 1×10^{10} . These findings provide compelling evidence of the pronounced therapeutic safety and efficacy in tumor suppression associated with the infused of 1-2C TCR-T cells.

In general details as to the number of times experiments were performed are missing. Similarly, statistical analysis is missing in most data.

Reply: Thank you for this reminder and we have added the number of times that the experiments were performed and the statistical analysis.

Reviewer #3 - Cancer immunotherapy, ACT - (Remarks to the Author):

The study by Lu and colleagues describes the generation of mutant KRAS-specific TCRs restricted to the HLA-A*11:01 in HLA-transgenic mice. KRAS G12V-specific T cells were labelled with tetramers and sorted from mouse splenocytes. The TCRs were isolated and sequenced before cloning into lentiviral expression vectors. The TCRs were evaluated in both CD8+ Jurkat cells and primary T cells.

Two of the TCRs that were well expressed and specific were selected for further study and tested against four cell lines transduced with HLA-A*11:01 and one also with different KRAS mutations.

The authors have very thoroughly evaluated the TCRs, studying functionality of transduced T cells in vitro and in xenograft models, as well as looking at TCR affinity and pMHC stability and the structure of TCR and pMHC. The study is very well conducted and the conclusions supported by the data. However, some of the data and methods require some clarification.

1. In Figure 1 the tetramer staining of TCR-expressing HEK-293T cells. The transduction efficiency was variable for the different TCRs and two TCRs were chosen for further study. In Figure 2 the HEK-293T cells expressing the two TCRs are assessed by tetramer staining for the 7 KRAS mutated 9-mer peptides and the corresponding wt peptide. Do all peptides bind the HLA-A*11:01 molecules equally well?

Reply: We agree that whether the mutant peptides could be loaded properly with HLA molecules are the basis for these binding assays. We therefore added the gel-filtration analysis of the wild type and mutated KRAS pHLA proteins as revised Figure S4. All the peptide could properly bind the HLA-A*11:01 molecule and therefore we believe that the tetramer staining results were reliable. The results from SPR binding analysis and functional studies also support the binding specificity of the TCRs identified in the present study.

2. In Figure S3 the intracellular cytokine production of TCR transduced T cells is assessed in response to PANC-1 peptide loaded target cells. Primarily CD8 T cells

respond, but the response rate is very low (2.85% IFN-g+ for I-2C TCR and 1.9% for 3-2E). What was % of TCR-expression in the evaluated T cells? Is there additional evidence for the CD8 dependency of the TCRs? The transduction efficiency also seemed to be quite low for the two TCRs in certain donors (Figure S4).

Reply:

(1) The percentages of TCR-expression in the evaluated T cells varied in different donors and were provided as revised Figure S6. Thank you for your reminder that we have added the cell source in the figure Legends.

(2) For CD8 dependency of the TCR, we have performed intracellular cytokine staining upon stimulation with corresponding peptides. Besides, we have also performed CD4/8 depletion ELISPOT assay and the results were added in revised Figure S5. In this experiment, the CD8+ T cells were isolated with anti-CD8 beads and the rest of the TCR-T cells considered as CD8- T cells were co-cultured with target cells to investigate CD4/8 dependency of the TCR-T cells. The IFN- γ -ELISPOT analysis revealed that no substantial responses could be observed for both 1-2C and 3-2E TCR-T cells when CD8+ TCR-T cells were depleted. All these results indicate that the immune responses observed for both 1-2C and 3-2E TCR-T cells were CD8 dependent.

(3) Individual variations in TCR expression have been observed in PBMCs derived from different donors following lentivirus infection. It is indisputable that higher transduction efficiency will bring better evaluation results. The current transduction efficiency has been optimized by our group after the experimental method, which can ensure the reliable verification of cellular functionality in subsequent analyses.

3. PANC-1 cells were modified to express different KRAS mutations. Their expression was assessed by flow cytometry (Figure S4), however the results are not shown and the details of the flow cytometry not outlined. Which anti-HA antibody was used, were the cells sorted as the expression of the different constructs was variable in the cell line?

Reply: In this study, we used an anti-HA tag monoclonal antibody (mAb) that was

purchased from MBL (code:No. M180-3, clone: TANA2). Our laboratory utilized puromycin selection to establish stable overexpressing cell lines, and we have included a comprehensive description of the culture conditions in the “Target cell preparation” of the materials and methods section.

4. In Figure 3 IFN-g ELISPOT assays were used to test both TCR-transduced T cells and assess the frequency of peptide-specific T cells in the spleen of immunized mice (Fig. S6). The ELISPOT assay for human T cells is fairly well described with reference to previous papers, however, it is not explained why the spot counts are expressed as number of spots per 10e6 cells as 10e5 T cells were plated per well.

The y-axis is labelled with TCR+ cells, does this mean the percentage of TCR expression was taken into account for the E:T ratios or does this refer to total number of T cells?

Was the expression level of the mutated KRAS the same for the PANC-1 cells (see point 3) I n Figure 3 g and j)?

The ELISAs show a clear dose-dependent response to the KRAS G12V peptide, with a higher affinity for the 1-2C TCR consistent with the SPR data presented in Figure 2.

Reply:

- 1) We agree with this suggestion and changed the y axis to SFCs per 10⁵ T cells.**
- 2) We did not take the percentage of TCR expression was taken into account and these were total TCR-T cells. We have also described this in the revised materials and methods section.**
- 3) The expression levels of mutated KRAS were completely the same for the PANC-1 cells, as is also showed in Supplementary Figure S6. This is a qualitative experiment to address the specificity of TCR-T cell responsiveness. The IFN- γ -ELISA assay with varied peptide concentrations has showed dose dependent and specific responses against KRAS-G12V. Together with the binding analysis and functional analysis, all these results support that the 1-2C and 3-2E TCRs were specific for KRAS-G12V in the context of HLA-A11.**

4) **The observed variation between 1-2C and 3-2E may potentially be attributed to differences in their binding affinity. 3-2E binds to KRAS-G12V-9/HLA-A*11:01 with KD values of $28.0 \pm 1.9 \mu\text{M}$, which is lower than that of 1-2C with $14.0 \pm 0.8 \mu\text{M}$ as shown in Figure 2. However, the antigen sensitivity of these two TCRs were similar in IFN-r-ELISA assay. Therefore, we speculate that 1-2C TCR may possess superior potency over 3-2E but still need further evaluations, which is not the focus of the present study.**

5. In Figure 4 where cross-reactivity to peptides with similar sequences was tested, donor 1, 2 3 are also used. Were these the same as the donors shown in Figure 3 and S4? Which peptide concentration was used in the assay presented?

Were any other methods or additional cytokines (TNF-a, IL-2) or degranulation markers (CD107a) used to verify cross-reactivity? How about other cell lines or healthy tissues?

Reply:

(1) These three donors were the same with the test in Figure 3. We have changed this figure to Supplementary Fig. S9 and added the T cell source in the figure legend. We did not test additional cytokines except for IFN- γ . The peptide concentration used for IFN- γ -ELISA was 10 $\mu\text{g}/\text{mL}$.

(2) As is also asked by Reviewer 1 and Reviewer 2, the specificity is a critical issue for the TCRs. As is also replied above, we performed tests against a combinatorial peptide library to address this concern. A combinatorial peptide library was synthesized with each of the residues substituted with 20 amino acids. The responses of 1-2C and 3-2E TCR-T cells constructed from three donors were tested with IFN- γ -ELISA assay. Besides, additional cytotoxicity analysis was also performed with PANC-1 and SW620 target cells and *in vivo* tumor suppression efficacy was additionally investigated for PANC-1 bearing mouse model. All these results indicated that 1-2C TCR was highly competent with anti-tumor efficacy with specific recognition and responses against tumor cells with KRAS-G12V mutant in the context of HLA-A*11:01.

6. Very few details about the ELISPOT used to test the murine T cells after immunization are provided. This cannot be the same kit, please provide details of the kit and cell numbers used (only E:T ratio is provided. Were conditions tested in triplicate? Only one well per condition is shown. The positive control (PMA/ionomycin) did not work well in all cases, please comment.

Reply: The ELISPOT used for murine T cells were different from that used for TCR-T cells and this has been changed in Materials and Methods section. The specific number of cells utilized in this experiment has been added in the figure legends of Figure S12. To ensure reliable negative and positive control, we opted not to replicate the experimental groups due to insufficient spleen cell counts in some mice. The less amount of survived human T cells may be the reason for the weak ELISPOT assay with spleen cells in some of the mice.

7. In vivo: NCG mice are used with a human xenograft (SW620) and human T cells presumable. However, the heading in the methods says syngeneic tumor model. Please correct.

The in vivo results in Figure 5 show potent anti-tumour reactivity of the I-2C T cells. How many times were T cells tested in NCG mice?

Anti-PD-1 treatment was also included in the in vivo experiment and showed some synergy with the T cell treatment, even in the group treated with mock T cells + anti-PD-1. Apart from one mouse in this group with a higher tumour load, the anti-PD-1 treatment was as efficacious as TCR transduced T cells without anti-PD-1. This should be discussed. Was PD-1 expression detected on the T cells? Did the SW620 cells express PD-L1?

Reply:

Thank you for your correction and we have corrected accordingly.

1) The present investigation comprises a tripartite in vivo evaluation, with the measurement of tumor size being presented in Figure 5, and another independent trial of luciferase measurement, conducted in human xenograft

(SW620 expressing HLA-A*11:01 and luciferase) mouse models, which was added in supplementary Figure S11. These assessments are critical in unraveling the repeatability of experiment and therapeutic efficacy of 1-2C TCR.

- 2) We performed additional *in vivo* tumor suppression experiment in PANC-1 tumor bearing NCG mouse model before the investigation in SW620 tumor model. 1-2C TCR-T cells were constructed with the PBMCs from two donors and three varied dose groups, high, media and low doses with 1×10^7 , 1×10^6 and 1×10^5 TCR-T cells, respectively, were infused. The results showed that substantial anti-tumor efficacy could be observed for high dose group and media dose group, whereas no substantial tumor suppression efficacy could be observed in low dose group. These results were added in revised Figure 5.
- 3) We agree that the tumor volumes in mock T cell and anti-PD-1 treatment group is lower than that of mock T cells, but did not reach statistical difference. We therefore changed the description in the manuscript as “Apart from one mouse in mock T and anti-PD-1 antibody co-treatment group with an exceptional higher tumor load, the tumor volumes in this group were substantially lower than that in mock T cell treatment group and comparable to that in TCR-T treatment group without anti-PD-1 antibody.”
- 4) As this reviewer suggested, we tested the surface expression of PD-1 in TCR-T cells and PD-L1 in SW620 tumor target cells, and the results were presented as revised Figure S11. The T-cell surface exhibited substantially upregulated expression of PD-1 in TCR-T cells. Surface expression of programmed death-ligand 1 (PD-L1) on SW620 tumor cells was substantially increased after co-culturing with TCR-T cells.

ELISPOT assays were carried out to assess the functional response of splenocytes in mice sacrificed on day 26. Was the TCR expression of the splenic T cells also assessed? Why were T cells from the TCR/anti-PD-1 treated group not tested in IFN- γ ELISPOT?

Why were mice sacrificed on day 26? Was survival assessed in any of these mice according to humane endpoints?

Reply:

(1) We have tested the responses in representative mice and found that only few T cells can be detected in spleen cells, and the TCR positive T cells cannot be detected on Day 26.

(2) The responses of two mice from TCR-T/anti-PD-1 co-treated group was tested and the results showed no substantial specific responses. Given the limitations of the data obtained, the corresponding figure has been relocated to the Supplementary Figure S12 of the revised manuscript to ensure greater clarity.

(3) On day 26, tumor sizes in the control group have reached an oversize requiring euthanasia and we did not follow the experiment beyond 26 days. We tried to analyze the T cell responses and tumor weight in all the groups, and therefore all the mice were sacrificed for comparative analysis. Additional experiment has performed to extend the follow-up duration with survival curve has been presented in Supplementary Figure S11.

8. The structural data in Figure 6 are interesting and the increased thermostability (Figure S9) of the KRAS-G12V -9/HLA-A*11:01 complex compared to the complex with wt KRAS peptide would be important for KRAS-specific TCR-based therapy.

Reply: Thank you for your comments. We agree that the thermostability analysis indicates that G12V may be preferably presented by HLA-A*11:01 than the other mutants. We therefore moved this result to Figure 7 for the highlight of this point.

9. The alanine scan performed in Figure 7 indicated that virtually all peptide positions were important for a functional T cell response. The alanine replacements could also disrupt the binding of the peptide to HLA-A*11:01, was this assessed in any way?

Reply: We have also analyzed the formation of pHLA complex of alanine substituted peptide by gel-filtration and the results has been presented as below. As is expected, the alanine substitution at the K9 position, which serves as primary

anchor residue for HLA-A*11:01 restricted peptides, has substantially attenuated the formation of pHLA complex. On the other side, the alanine substitutions at the other position did not affect the formation of stable pHLA complex.

However, because we have tested the cross responses against the combinatorial peptide library, as is also replied for Q5 and for Reviewer 2, we decided to remove this part from the revised Figure 7. The results for peptide library are more comprehensive and have covered the responses of alanine scanning.

10. The English language needs editing in parts of the paper. Some sentences need grammatical improvement and some wording needs changing (e.g. mice were sacrificed rather than slaughtered).

The abstract should be rewritten and the conclusions could be bolder regarding future prospects of TCR-based therapy.

Reply: The abstract has been rewritten in the revised manuscript. The manuscript was re-edited thoroughly and has been edited by a native English speaker.

REVIEWER COMMENTS

Reviewer #1 (Remarks to the Author):

The authors have responded satisfactorily to the previous critiques.

Reviewer #2 (Remarks to the Author):

The authors have addressed most of the issues raised by this reviewer. However, two points still require further analysis / information.

First, authors are commended for performing the peptide combinatorial library analysis for evaluation of TCR cross-reactivity. This information is critical regarding the relevance of these TCRs for potential therapeutic application. Details on how the data derived from analysis was evaluated to arise at the 16 peptides listed in the supplementary data S9 is missing. What method was used to identify these non-cognate peptides? Use of the Scanprosite website is commonly used for identification (please see PMID: 28645940). This analysis appears to be missing.

Second, comments regarding in vivo data are not addressed. Is there a survival advantage on experiments presented on Fig 5c-g? How about Fig 5g-o? An extended monitoring of mice beyond 24 days and the reporting on survival is customary.

Reviewer #3 (Remarks to the Author):

The authors have addressed most of my comments and concerns sufficiently.

There is just one issue to clarify:

In the in vivo experiments shown in the original Figure 5, the authors were asked why the mice were sacrificed on day 26. In point 7, the authors reply that this was due to high tumour burden.

An additional experiment has been included in supplementary Figure 11 including a survival curve.

This experiment is very similar to the one presented in the Figure 5 for the SW620 cells. In

the first experiment the mice had to be euthanized on day 26 due to high tumour load in the control group, however, in Supplementary Figure 11, the control mice are euthanized just before day 50 and the treated mice survive much longer (cut-off at day 80). The measurement of tumour load is done differently in the two experiments; with caliper in Figure 5 and by IVIS in Supplementary Figure 11, and the survival is nearly twice as long for the control mice in Supplementary Figure 11. What was the reason for this?

REVIEWER COMMENTS

Reviewer #2 (Remarks to the Author)

The authors have addressed most of the issues raised by this reviewer. However, two points still require further analysis / information.

First, authors are commended for performing the peptide combinatorial library analysis for evaluation of TCR cross-reactivity. This information is critical regarding the relevance of these TCRs for potential therapeutic application. Details on how the data derived from analysis was evaluated to arise at the 16 peptides listed in the supplementary data S9 is missing. What method was used to identify these non-cognate peptides? Use of the Scanprosite website is commonly used for identification (please see PMID: 28645940). This analysis appears to be missing.

Reply:

Thanks for the suggestion of the peptide combinatorial library analysis and the results have indeed substantially improved the characterization of recognition specificity of the TCRs. We believe this is a more comprehensive study than the previous results regarding the responses against 16 homologous peptides, and was therefore added as Figure 3. The method used for the selection of homologous peptides in human genome is based on Protein Blast (<https://blast.ncbi.nlm.nih.gov/Blast.cgi>) ((Supplementary Fig. S9). Homologous peptides were screened against whole genome of Homo sapiens (taxid:9606) and peptides shared more than six out of the nine residues with KRAS-G12V-9 peptide (66.7% similarity) were selected for analysis. The details of peptide selection were added in the revised manuscript.

We further use Scanprosite to analyze homologous peptides, as this reviewer suggested. The recognition motifs of 1-2C and 3-2E TCRs were different based on the results from peptide combinatorial library analysis, and therefore different motifs were used to identify homologous peptides for 1-2C and 3-2E TCRs. [GAVIKMCHR]-[GVIFWNQ]-G-[ASC]-[VM]-G-[VI]-G-[KR] motif was used for 1-2C TCR, and [VI]-[GVWDST]-G-[ALC]-[VMH]-G-[VI]-[GAFYQMSHR]-[KR] motif for 3-2E TCR. Three homologous peptides were identified for 1-2C TCR recognition motif, (UniProt: Q6H795, RWGAVGVGR; UniProt: 72WD6, IFGSVGVGK; Uniprot: P9WFD4, CVGSVGIGR), whereas two homologous peptides were identified for 3-2E TCR (UniProt: Q2GC58, VSGCVGVFR; UniProt: A0PXA8, ITGAVGIAK). However, none of these homologous peptides were from human genome. These analyses have been added in revised manuscript.

Taken together, we believe that the data presented in the revised manuscript could support the specificity characterizations of the reported TCRs.

Second, comments regarding in vivo data are not addressed. Is there a survival advantage on experiments presented on Fig 5c-g? How about Fig 5g-o? An extended

monitoring of mice beyond 24 days and the reporting on survival is customary.

Reply: The mice in each experimental groups were sacrificed on day 16 or day 26 for PANC-1 or SW620 model, respectively, for a parallel comparison of the anti-tumor efficacy in different treatment groups. Indeed, 1-2C TCR-T showed significant antitumor effect according to the tumor growth curve. To further investigate whether 1-2C TCR-T also has a survival advantage, we therefore performed additional experiments in SW620 model (Supplementary Figure S11) as suggested by this reviewer in the previous revision. The monitoring of tumor burden was extended to day 74 and 1-2C TCR-T treated group showed substantial survival advantage compared with control groups. The more detailed information about this IVIS imaging methods were provided in Materials and Methods section.

Therefore, we believe the data presented here could support the tumor suppression potency of the 1-2C TCR-T cells. The concerns of the survival advantage in PANC-1 and SW620 models presented in Figure 5 were added in results section and the survival data were described accordingly in revised manuscript. The endpoint criteria of animal experiments were also provided in Materials and Methods section.

Reviewer #3 (Remarks to the Author)

The authors have addressed most of my comments and concerns sufficiently.

There is just one issue to clarify:

In the in vivo experiments shown in the original Figure 5, the authors were asked why the mice were sacrificed on day 26. In point 7, the authors reply that this was due to high tumour burden.

An additional experiment has been included in supplementary Figure 11 including a survival curve.

This experiment is very similar to the one presented in the Figure 5 for the SW620 cells. In the first experiment the mice had to be euthanized on day 26 due to high tumour load in the control group, however, in Supplementary Figure 11, the control mice are euthanized just before day 50 and the treated mice survive much longer (cut-off at day 80). The measurement of tumour load is done differently in the two experiments; with caliper in Figure 5 and by IVIS in Supplementary Figure 11, and the survival is nearly twice as long for the control mice in Supplementary Figure 11. What was the reason for this?

Reply:

The additional experiment in Supplementary Figure 11 was performed as suggested by Reviewer #2 to extend mice monitoring beyond 24 days and to analyze survival advantage of TCR-T cell treatment. For a more sensitive and objective measurement of tumor burden, SW-620-A11-luci tumor cells (SW620 expressing HLA-A*11:01 and luciferase) were constructed and IVIS imaging were used for tumor monitoring in revised Supplementary Figure 11, whereas SW-620-

A11 tumor cells (SW620 expressing HLA-A*11:01) were used in Figure 5 for tumor weights and tumor volume monitoring. The SW620-A11 and SW620-A11-luci used in Figure 5 and Supplementary Figure 11, respectively, are two different subclones and may explain the difference in tumor burden between the two sets of experiments. The more detailed information about this IVIS imaging methods were provided in Materials and Methods section. The endpoint criteria of animal experiments were also provided.

The mice in each experimental groups were sacrificed on day 26 for SW620-A11 model for a parallel comparison of tumor weights in Figure 5 and therefore the data of survival advantages were not obtained. We have noticed the concern raised by Reviewer #2 and therefore kept the mice for survival monitoring in Supplementary Figure S11.

REVIEWERS' COMMENTS

Reviewer #2 (Remarks to the Author):

No additional comments. The authors have addresses all points raised by reviewer.